# Variability in the Munc13-1 content of excitatory release sites

**Maria Rita Karlocai, Judit Heredi, Tünde Benedek, Noemi Holderith, Andrea Lorincz, Zoltan Nusser***

Laboratory of Cellular Neurophysiology, Institute of Experimental Medicine, Budapest, Hungary

**Abstract** The molecular mechanisms underlying the diversity of cortical glutamatergic synapses are still incompletely understood. Here, we tested the hypothesis that presynaptic active zones (AZs) are constructed from molecularly uniform, independent release sites (RSs), the number of which scales linearly with the AZ size. Paired recordings between hippocampal CA1 pyramidal cells and fast-spiking interneurons in acute slices from adult mice followed by quantal analysis demonstrate large variability in the number of RSs ($N$) at these connections. High-resolution molecular analysis of functionally characterized synapses reveals variability in the content of one of the key vesicle priming factors – Munc13-1 – in AZs that possess the same $N$. Replica immunolabeling also shows a threefold variability in the total Munc13-1 content of AZs of identical size and a fourfold variability in the size and density of Munc13-1 clusters within the AZs. Our results provide evidence for quantitative molecular heterogeneity of RSs and support a model in which the AZ is built up from variable numbers of molecularly heterogeneous, but independent RSs.

## Introduction

Computational complexity of neuronal networks is greatly enhanced by the diversity in synaptic function (*Dittman et al., 2000*; *O'Rourke et al., 2012*). It has been known for decades that different types of central neurons form synapses with widely different structures, molecular compositions, and functional properties, resulting in large variations in the amplitude and kinetics of the postsynaptic responses and the type of short- and long-term plasticity. When the mechanisms underlying distinct functions were investigated among synapses made by distinct pre- and postsynaptic cell types (e.g. hippocampal mossy fiber vs. Schaffer collateral vs. calyx of Held vs. cerebellar climbing fiber etc. synapses), most studies converged to the conclusion that different pre- (e.g. different types of Ca channels, Ca sensors) and postsynaptic (e.g. different types of AMPA receptor subunits) molecule isoforms underlie the functional variability (reviewed by *Südhof, 2012*).

Robust differences in synaptic function were also found when a single presynaptic cell formed synapses on different types of postsynaptic target cells. Such postsynaptic target cell type-dependent variability in vesicle release probability ($Pv$) and short-term plasticity was identified in cortical and hippocampal networks (*Koester and Johnston, 2005*; *Losonczy et al., 2002*; *Pouille and Scanziani, 2004*; *Reyes et al., 1998*; *Rozov et al., 2001*; *Scanziani et al., 1998*; *Thomson, 1997*). Studies investigating the underlying mechanisms revealed not only different molecules (e.g. mGluR7, kainate receptors in the AZ and Elfn1 in the postsynaptic density [PSD], *Shigemoto et al., 1996*; *Sylwestrak and Ghosh, 2012*), but variable densities of the same molecules were also suggested as key molecular features (*Éltes et al., 2017*; *Rozov et al., 2001*).

Probably even more surprising is the large structural and functional diversity of synapses that are established by molecularly identical pre- and postsynaptic neuron types (e.g. synapses among cerebellar molecular layer interneurons [INs], *Pulido et al., 2015*; among hippocampal CA3,

**\*For correspondence:**
nusser.zoltan@koki.mta.hu

**Competing interests:** The authors declare that no competing interests exist.

*Holderith et al., 2012*), suggesting that qualitative molecular differences are unlikely to be responsible for the functional diversity. What could then be responsible for the large diversity in function in such synapses? *Pulido et al., 2015* investigated so called simple synapses where the synaptic connection is mediated by a single presynaptic AZ and the opposing PSD. Their results revealed that the number ($N$) of functional release sites (RSs) varied from 1 to 6 per AZ, and it showed a positive correlation with the quantal size ($q$). Because in these synapses $q$ is largely determined by the number of postsynaptic GABA$_A$ receptors and because the GABA$_A$ receptor number scales linearly with the synapse area (*Nusser et al., 1997*), they concluded that the $N$ linearly scales with the synaptic area. Previous results from our laboratory showed that the probability with which release occurs ($P_R$) from a CA3 PC axon terminal correlates with the size of the synapse. As this probability is the function of both $Pv$ and $N$ [$P_R = 1 - (1 - Pv)^N$], our results are also consistent with the model that $N$ scales with the synaptic area (*Holderith et al., 2012*). This view was further supported by a recent paper (*Sakamoto et al., 2018*), which concluded that in synapses of cultured hippocampal neurons the number of Munc13-1 macromolecular clusters shows a linear correlation with the $N$ and no correlation with $Pv$. Since Munc13-1 and its invertebrate homologs are essential for docking and priming of synaptic vesicles (*Augustin et al., 1999*; *Imig et al., 2014*; *Weimer et al., 2006*) and the fact that its cluster numbers show tight correlation with $N$ suggests that it is a key molecule of the RSs and can be used as its molecular marker. Thus, the following model emerged: presynaptic AZs are composed from an integer number of uniform, independent RSs, which are built from the same number of identical molecules (molecular units). The more RSs there are, the larger the size of the AZ is, which faces a correspondingly larger PSD containing proportionally more receptors. This model is supported by a number of molecular neuroanatomical studies showing that the number of presynaptic AZ molecules (e.g. Cav2.1, Cav2.2, Rim1/2; *Holderith et al., 2012*; *Kleindienst et al., 2020*; *Miki et al., 2017*) or postsynaptic molecules (e.g. PSD-95, AMPA receptors; *Fukazawa and Shigemoto, 2012*; *Kleindienst et al., 2020*) scales linearly with the synapse area. Furthermore, it was also suggested that each RS faces a cluster of postsynaptic AMPA receptors in a so called nanocolumn arrangement (*Tang et al., 2016*). However, a recent study using superresolution imaging of release from cultured neurons concluded that the RSs are functionally heterogeneous and RSs with high or low $Pv$ are distributed in a nonrandom fashion within individual AZs (*Maschi and Klyachko, 2020*).

Here, we performed in vitro paired whole-cell recordings followed by quantal analysis to determine the quantal parameters ($N$, $Pv$, and $q$) in synaptic connections between hippocampal CA1 pyramidal cells (PCs) and fast-spiking interneurons (FSINs). Our results demonstrate that the large variability in postsynaptic response amplitude is primarily the consequence of large variations in $N$. The variability in $N$ is also substantial in individual AZs ($N$/AZ: 1–17). Multiplexed molecular analysis with STED superresolution microscopy revealed large variability in the Munc13-1 content of AZs that possess the same $N$, indicating that RSs could be formed by variable number of Munc13-1 molecules. This molecular variability among RSs is supported by our high-resolution electron microscopy replica immunolabeling data, demonstrating highly variable number of gold particles in Munc13-1 clusters in these hippocampal glutamatergic AZs. Whether vesicles that are docked to RSs with different amounts of Munc13-1 have distinct $Pv$ or not remains to be seen.

## Results

### Large variability in unitary EPSC amplitudes evoked by CA1 PCs in FSINs

To investigate the variance in unitary EPSC (uEPSC) amplitudes evoked in FSINs by CA1 PC single action potentials (APs), we recorded a total of 79 monosynaptically connected pairs in 2 mM external [Ca$^{2+}$] from acute slices of adult mice of both sexes (*Figure 1*). The amplitude of uEPSCs ranged from 3 to 507 pA with a mean of 105.0 pA and a SD of 107.9 pA, yielding a coefficient of variation (CV) of 1.03. The uEPSCs had a moderate variability in their 10–90% rise times (RT, mean = 0.4 ± 0.2 ms, CV = 0.4), but some had values over 1 ms. To exclude the contribution of differential dendritic filtering to the observed variance in amplitudes, we restricted our analysis to presumed perisomatic synapses by subselecting cells with mean uEPSC 10–90% RTs ≤ 500 µs. These fast-rising EPSCs had a similar large variability in their amplitudes (113.1 ± 111.0 pA, n = 68; *Figure 1D*), with a CV of 0.98. The type of short-term plasticity is a widely used feature of postsynaptic responses that is

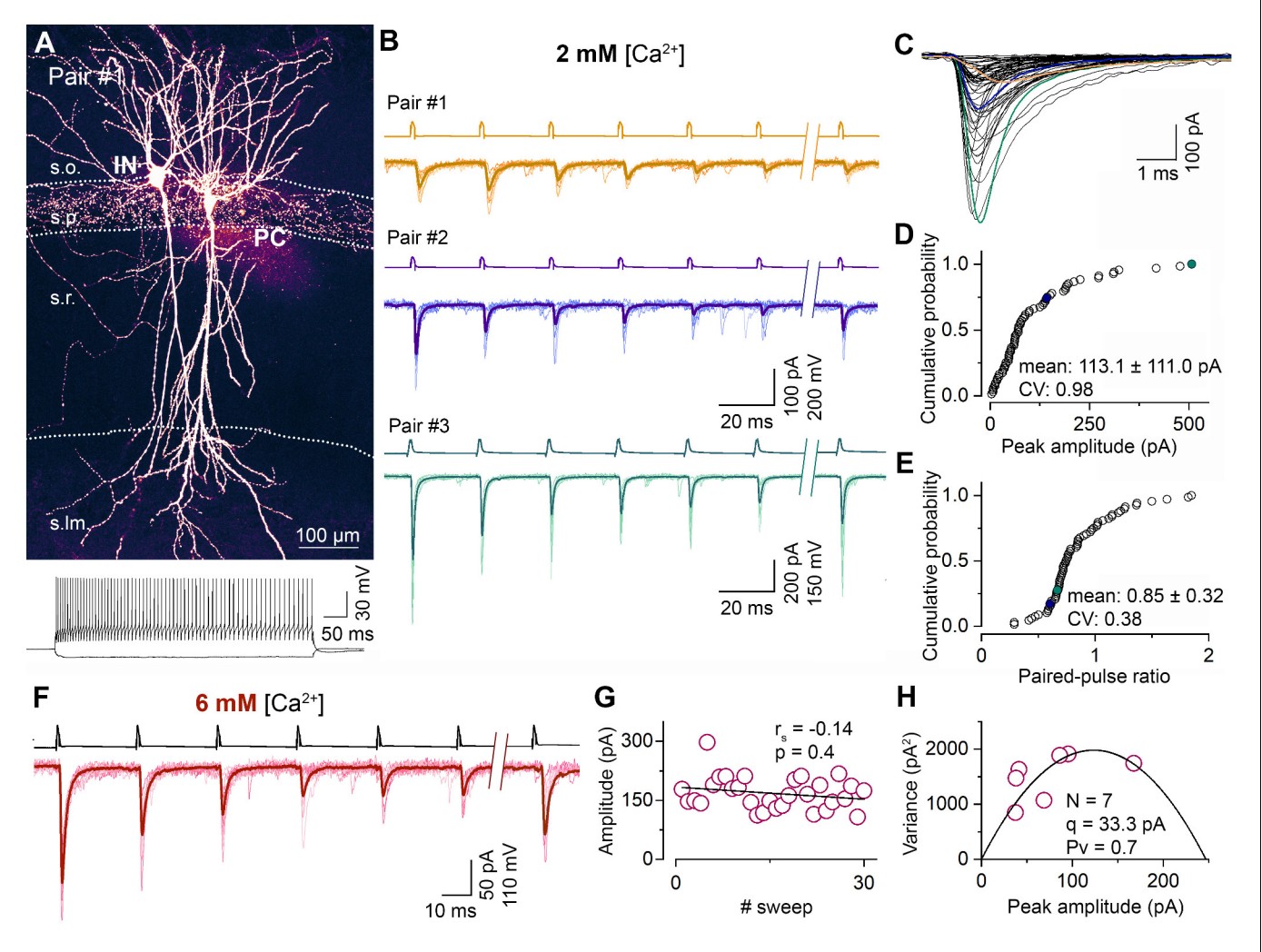

**Figure 1.** Synapses between CA1 PC and FSINs show a large variability in their postsynaptic response amplitude and short-term plasticity. (**A**) Representative confocal image of a monosynaptically connected, biocytin labeled PC–FSIN pair in the hippocampal CA1 region (top). Membrane potential responses of the IN upon depolarizing and hyperpolarizing current injections (bottom). The supratreshold response shows FS firing characteristics. (**B**) Excitatory connections in three PC–FSIN pairs. EPSCs (lower traces) recorded in postsynaptic FSINs evoked by action potential (AP) trains in the presynaptic PCs (6 APs at 40 Hz followed by a recovery pulse at 300 or 500 ms, upper traces) display large variability in amplitude and short-term plasticity in 2 mM $[Ca^{2+}]$. (Pair #1, 68.0 ± 27.7 pA, PPR: 1.05; Pair #2, 140.7 ± 50.8 pA, PPR: 0.58; Pair #3, 507.4 ± 199.6 pA, PPR: 0.65). Top scale bars apply to the top two traces. Recording shown in orange (Pair #1) are from the cell pair in (**A**). (**C**) Superimposed averaged traces of the first EPSCs (n = 65 from 50 mice, mean EPSC rise time: 0.4 ± 0.2, CV = 0.40). Colored traces are from the corresponding pairs shown in (**B**). (**D, E**) Cumulative distribution of the peak amplitudes of the rise time-subselected first EPSCs and the paired-pulse ratios (EPSC2/EPSC1) recorded in 2 mM $[Ca^{2+}]$ (n = 68 pairs from 46 mice; mean ± SD are indicated on the figure). Colored symbols represent two corresponding pairs with rise times ≤ 0.5 ms shown in (**B**). (**F**) Unitary EPSCs in a representative PC–FSIN pair recorded in 6 mM $[Ca^{2+}]$. Same stimulation protocol as in (**B**). (**G**) Stability of the peak amplitude of the first EPSCs over 30 sweeps from the pair shown in (**F**). (**H**) Relationship between mean and variance values of EPSC peak amplitudes in 6 mM $[Ca^{2+}]$ from the pair shown in (**F**). Quantal parameters were estimated with MPFA. *N*, number of functional release sites, *q*, quantal size, *Pv*, vesicular release probability. $r_s$, Spearman's rank correlation coefficient. s.o. stratum oriens, s.p. stratum pyramidale, s.r. stratum radiatum, s.lm. stratum lacunosum-moleculare.

The online version of this article includes the following source data and figure supplement(s) for figure 1:

**Source data 1.** Synapses between CA1 PC and FSINs show a large variability in their postsynaptic response amplitude and short-term plasticity.

**Figure supplement 1.** Confocal and STED analysis of Munc13-1 and PSD-95 immunosignals at functionally characterized synapses in a CA1 PC–FSIN pair that is shown in *Figure 1F–H*.

assumed to predict the *Pv*. Although some connections displayed initial facilitation followed by depression, most of the connections showed robust depression, and the resulting moderate variability in the paired-pulse ratio (CV = 0.38; *Figure 1E*) implies that the variability in *Pv* might not be the major source of variability in EPSC amplitudes. It is well known that FSINs are morphologically diverse (contain perisomatic region-targeting basket and axo-axonic cells and dendrite-targeting bistratified cells) and therefore we tested whether the observed amplitude variance could be the consequence of different morphological identity of the postsynaptic cells. A total of 50 FSINs could be categorized into perisomatic region-targeting (n = 35) or bistratified (n = 15) cells, and when uEPSCs amplitudes were compared (perisomatic: 128.1 ± 121.9 pA vs bistratified: 126.4 ± 125.7 pA), no significant difference was found (p=0.98, Mann–Whitney U-test). Furthermore, the CV within each group was ~1, revealing a similar variance in EPSC peak amplitudes when the postsynaptic cells belong to a well-defined IN category.

## Quantal parameters at PC–FSIN connections

To elucidate the basis of the uEPSC amplitude variability, we determined the quantal parameters *N*, *Pv*, and *q* of the connections using multiple-probability fluctuation analysis (MPFA, *Silver, 2003*). For their reliable determination, the *Pv* must be changed substantially and must have a maximum value > 0.5. We aimed to achieve these by elevating the external $[Ca^{2+}]$ to 6 mM and applying a train of presynaptic APs (6 APs at 40 Hz) within which the *Pv* changes dynamically (*Biró et al., 2005*; *Figures 1F–H*). We also bath applied the CB1 receptor antagonist AM251 to increase further the *Pv* (the effect of AM251 in separate experiments: control: 68.0 ± 16.5 pA; AM251: 78.0 ± 23.2 pA, n = 5 pairs) and to minimize potential variability due to differential presynaptic tonic CB1 receptor activations. The peak amplitude of uEPSCs in 6 mM $[Ca^{2+}]$ was significantly higher (165.5 ± 169.3 pA, n = 100; p=4.42 × $10^{-4}$, Mann–Whitney U-test) than in 2 mM extracellular $[Ca^{2+}]$ but showed similarly large variability (CV = 1.0). The RT-subselected, presumably perisomatic uEPSCs had a mean amplitude of 183.4 ± 180.7 pA (n = 81) with a CV of 0.99, confirming our results in 2 mM $[Ca^{2+}]$ that dendritically unfiltered EPSCs are also highly variable. Of these 81 pairs, we managed to reliably determine the quantal parameters (see Materials and methods) in 47 pairs (peak amplitude: 215.8 ± 211.2 pA, CV = 0.98; *Figure 2A*) and found large variability in *N* (9.9 ± 9.0, CV = 0.91; *Figure 2B*), a much smaller variance in *q* (32.4 ± 16.0 pA, CV = 0.49; *Figure 2C*) and an especially low variance in *Pv* (0.72 ± 0.1, CV = 0.14; *Figure 2D*). Peak amplitude of uEPSCs correlated tightly with *N* (Spearman regression coefficient ($r_s$) = 0.79; *Figure 2E*), less tightly with *q* ($r_s$ = 0.38; *Figure 2F*) and with *Pv* ($r_s$ = 0.36; *Figure 2G*), demonstrating that variability in *N* is the major determinant of the uEPSC amplitude variability.

The small variance in *Pv* and its small contribution to the total amplitude variance in 6 mM $[Ca^{2+}]$ is not surprising given the ceiling effect of artificially increasing the release. To investigate its variance under more physiological $[Ca^{2+}]$, we recorded cell pairs in 2 mM, then subsequently in 6 mM $[Ca^{2+}]$ (*Figure 2—figure supplement 1*). The *Pv* was then determined with MPFA in 6 mM $[Ca^{2+}]$, and its value in 2 mM $[Ca^{2+}]$ was calculated from the uEPSC amplitude ratio, assuming that changing extracellular $[Ca^{2+}]$ only affects *Pv*. As expected, the *Pv* was smaller (mean = 0.42 ± 0.15, n = 14) and more variable (CV = 0.36) in 2 mM $[Ca^{2+}]$ when compared to that in 6 mM $[Ca^{2+}]$ (mean = 0.71 ± 0.10, CV = 0.14, n = 14). Because *Pv* in 2 mM $[Ca^{2+}]$ shows a more pronounced correlation with the peak EPSC amplitude (*Figure 2—figure supplement 1C*), we calculated the relative contribution of the three quantal parameters to the amplitude variance and found that even in 2 mM $[Ca^{2+}]$, the variance in *N* (63%) has a substantially larger contribution than that of *q* (25%) or *Pv* (12%; for CV values in 2 mM $[Ca^{2+}]$, see *Figure 2—figure supplement 1A*).

Because PC–FSIN connections are not mediated by single synapses (*Buhl et al., 1997*; *Molnár et al., 2016*), the overall variability in *N* is not simply the consequence of different N/AZ, but also the function of the number of synaptic contacts formed by the presynaptic axon on the postsynaptic cell. To determine the number of synaptic contacts between the connected cells, we carried out high-magnification confocal microscopy analysis of the biocytin-filled, aldehyde-fixed, and post hoc developed cells (detailed below). Our data revealed a relatively weak correlation between peak uEPSC amplitude and the N/AZ ($r_s$ = 0.37; *Figure 2H*) and a more robust one between the peak uEPSC amplitude and the synapse number ($r_s$ = 0.61; *Figure 2I*). When we examined their variances, an approximately equal contribution of the synapse number (mean = 2.3 ± 1.6, n = 26, CV = 0.68)

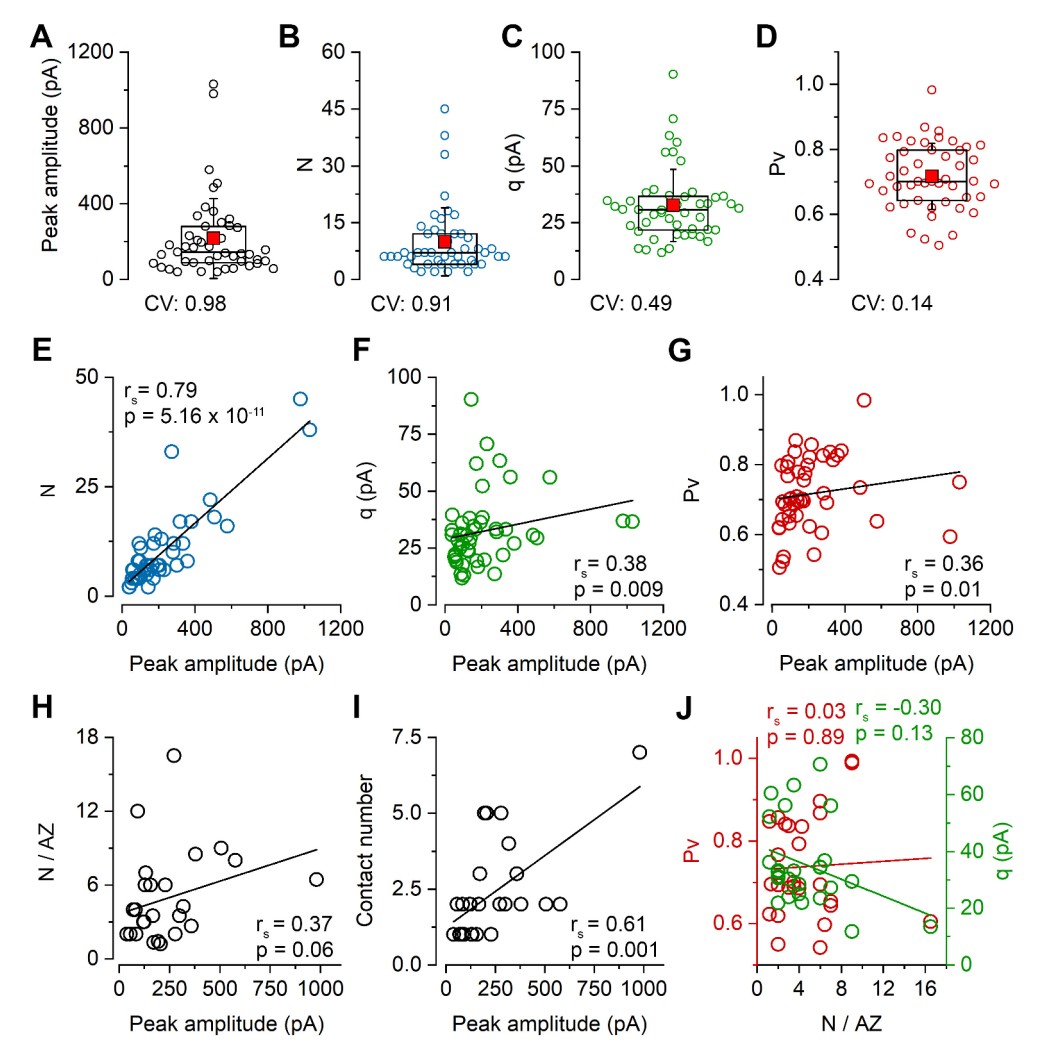

**Figure 2.** Variability in the number of release sites is primarily responsible for the variability in peak EPSC amplitudes at PC–FSIN pairs. (A–D) Distribution in the peak amplitude of the first EPSCs (mean: 215.9 ± 211.2 pA), the number of release sites ($N$, mean: 9.9 ± 9.0), quantal size ($q$, mean: 32.4 ± 16.0 pA), and vesicular release probability ($Pv$, mean: 0.72 ± 0.1) in 47 pairs from 41 mice in 6 mM [$Ca^{2+}$]. Boxplots represent 25–75% percentile, median (middle line), mean (red square), and SD (whisker) values. (E–I) Relationship between the peak amplitude of the first EPSC and the $N$ (E), $q$ (F), $Pv$ (G), N/AZ(H), number of anatomical contact sites (I) in 6 mM [$Ca^{2+}$], (E–G n = 47 pairs, H and I n = 61 contacts from 26 pairs, 25 mice). (J) Relationships between $Pv$ and N/AZ and $q$ and N/AZ are shown in 26 pairs. $r_s$, Spearman's rank correlation coefficient.

The online version of this article includes the following source data and figure supplement(s) for figure 2:

**Source data 1.** Variability in the number of release sites is primarily responsible for the variability in peak EPSC amplitudes at PC–FSIN pairs.

**Figure supplement 1.** Comparison of quantal parameters and their correlations in 2 mM and 6 mM extracellular [$Ca^{2+}$].

**Figure supplement 1—source data 1.** Comparison of quantal parameters and their correlations in 2 mM and 6 mM extracellular [$Ca^{2+}$].

and the N/AZ (mean = 4.9 ± 3.7, n = 26, CV = 0.75) to the variance in $N$ (mean = 10.2 ± 9.8, n = 26, CV = 0.96) was observed.

## Correlation of the amounts of synaptic molecules with *N*

So far, our results demonstrate large variability in the peak amplitude of uEPSCs between CA1 PC and FSINs, which is primarily the consequence of large variability in *N* among the connections. This variability originates approximately equally from differences in the number of synaptic contacts between the connected cells (ranges from 1 to 7, CV = 0.68) and from variations in *N*/AZs (ranges from 1 to 17, CV = 0.75). The mechanisms underlying the variability in the number of synaptic contacts between monosynaptically connected cells are unknown and addressing this issue is outside the scope of the present study.

Here we address the question of what the molecular correlates of the variability in the *N*/AZs are. We employed a recently developed high-resolution, quantitative, multiplexed immunolabeling method (*Holderith et al., 2020*) to molecularly analyze functionally characterized individual synapses. Following in vitro paired recordings, the slices were fixed, re-sectioned at 70–100 µm, the biocytin-filled cells were visualized with Cy3-coupled streptavidin and the sections were dehydrated and embedded into epoxy resin. As Cy3 molecules retain their fluorescence in the water-free environment of epoxy resins (*Figure 1A*, *Figure 1—figure supplement 1* and *Figure 3A–C*), we could search for potential contact sites between PC axons and the IN soma/dendrites under epifluorescent illumination using a high numerical aperture (NA = 1.35) objective lens. Every pair was studied by two independent investigators and all independently found potential contacts were scrutinized by three experts. After obtaining confocal image Z-stacks from all potential contacts, the sections were re-sectioned at 200 nm thickness, in which we could unequivocally identify the contacts (compare *Figure 3F,G*). Following their registration with confocal microscopy, multiplexed immunoreactions were carried out on serial sections for presynaptic Munc13-1, vGluT1 molecules and postsynaptic AMPA receptors (with a pan-AMPAR Ab, data not shown) and PSD-95 (*Figure 3H,I*, *Figure 1—figure supplement 1*). The reaction in each staining step was imaged with confocal and STED microscopy in each relevant serial section and following an elution step they were re-stained and re-imaged. The presence of vGluT1 immunoreactivity in the boutons and the opposing Munc13-1 and PSD-95 labeling at the sites of bouton–dendrite appositions were taken as evidence for the contacts being chemical glutamatergic synapses (*Figure 3H,I*, *Figure 1—figure supplement 1B,D*).

We then analyzed the amounts of PSD-95 and Munc13-1 molecules in the functionally characterized synapses quantitatively. We have chosen PSD-95 because its amount correlates almost perfectly with the size of the synapse (see Figure 5C and *Cane et al., 2014*; *Meyer et al., 2014*) and, therefore, we use it as a molecular marker of the synapse size and concentrated on Munc13-1 as it has been suggested to be a core component of the RS (*Reddy-Alla et al., 2017*; *Sakamoto et al., 2018*). Immunoreactivity for both molecules in the functionally characterized synapses was normalized to that of the population mean of the surrounding synapses, ruling out variations in our data due to slight differences in slice conditions, fixations, or immunoreactions. We have analyzed a total of 11 cell pairs: five had only one, five had two, and one had three synaptic contacts. The summed immunoreactivity for Munc13-1 showed a significant positive correlation with the number of contacts per connection (*Figure 3J*, left panel). For the single-contact connections, the functionally determined *N* and the amounts of molecules can be directly correlated. However, for the multi-contact pairs, it is unknown how the *N*s are distributed among the synapses. Based on the assumption that *N* positively correlates with the size of the synapses, we allocated the *N*s to the two or three synapses based on their size (assessed from their PSD-95 immunoreactions). When the correlations between the Munc13-1 immunoreactivity and *N*/AZ was examined, a significant positive correlation was found ($r_s$ = 0.57; *Figure 3J*, middle panel), but no significant correlation was observed between the Munc13-1 immunoreactivity and the *Pv* ($r_s$ = 0.02; *Figure 3J*, right panel).

Since *N* is the function of how many contacts there are between the cells and how many RSs there are within the AZs, we next dissected their individual contributions. Although our results revealed positive correlations for both values with Munc13-1 (*Figure 3J*), we noticed a remarkable variability: synapses with widely different *N*/AZ had similar amounts of Munc13-1 and synapses with similar *N*/AZ showed very different amounts of Munc13-1 (*Figure 3J*). In summary, our data are consistent with a model in which the size of the presynaptic AZ correlates with the *N*/AZ, but the observed variance indicates variability in the overall amounts of Munc13-1 in individual RSs. Next, we aimed to investigate this issue with a more sensitive and higher resolution method.

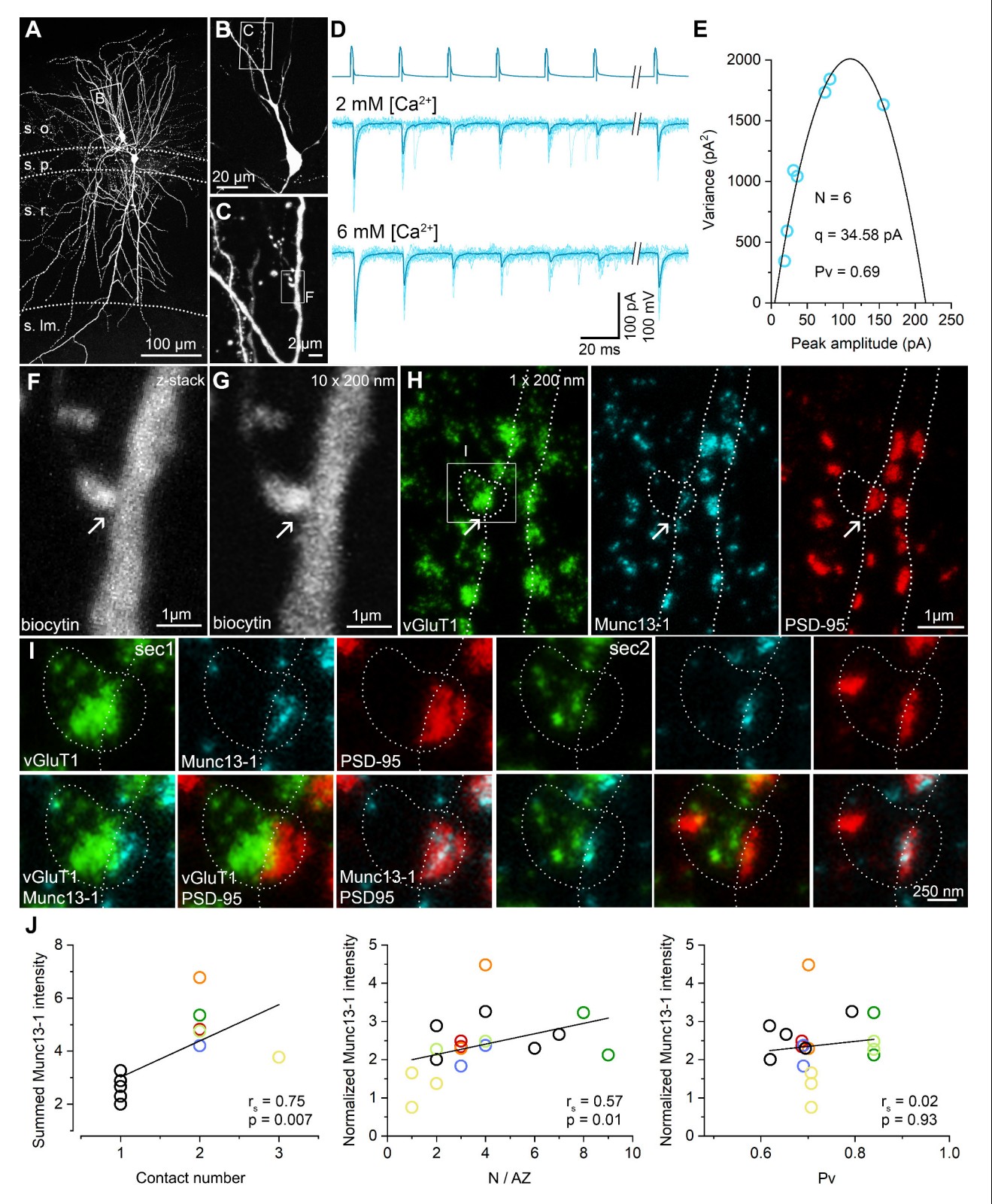

**Figure 3.** STED analysis of Munc13-1 and PSD-95 immunosignals at functionally characterized PC–FSIN synapses. (**A**) Confocal maximum intensity projection image of a monosynaptically connected, biocytin-filled PC–FSIN pair in the hippocampal CA1 region. (**B**) Enlarged view of the boxed area in (**A**) with the IN soma and proximal dendrites. (**C**) Confocal image stack enlarged from the boxed area in (**B**). Boxed region indicates the location of the synaptic contact site shown in (**F and G**). (**D**) Unitary EPSCs recorded from the pair shown in (**A**) in the presence of 2 mM or 6 mM [Ca²⁺]. Six APs were

*Figure 3 continued on next page*

*Figure 3 continued*

evoked at 40 Hz followed by a recovery pulse at 500 ms. (E) Relationship between mean and variance values of EPSC peak amplitudes in the presence of 6 mM extracellular [Ca$^{2+}$]. Quantal parameters were estimated with MPFA. (F) Maximum intensity projection image of confocal z-stacks (from seven optical sections at 300 nm steps) was obtained from a 125 µm thick resin-embedded slice. Arrow points to the putative synaptic contact between the PC axon and the IN dendrite. (G) The same contact is shown reconstructed from 10 thin (200 nm) serial sections after re-sectioning the resin-embedded slice. (H) STED microscopy image of a single 200 nm thin section. White dashed lines outlining the presynaptic bouton and the postsynaptic dendrite are superimposed on all images. Excitatory synapses – including the identified connection – are located along the biocytin filled dendrite identified by vGluT1 (green), Munc13-1 (cyan), and PSD-95 (red) triple immunolabeling. Arrows point to the putative synaptic contact between the PC axon and the IN dendrite. White box indicates the location of the enlarged area in (I). (I) Localization and separation of the presynaptic (vGluT1 and Munc13-1) and postsynaptic (PSD-95) proteins in the identified contact on two consecutive sections. The biocytin-filled bouton is labeled for vGluT1 (green). The close apposition of the Munc13-1 and PSD-95 immunosignals on the merged STED image confirms that the presynaptic axon forms a synapse on the postsynaptic dendrite. (J, left panel) Summed Munc13-1 intensity of synapses of each pair as a function of contact number. In pairs with double or triple contacts the sums of Munc13-1 intensities are plotted. Black circles represent single-contact pairs, colored circles represent pairs with two (red, blue, green, orange, light green) or three (yellow) synaptic contacts throughout the panels. The Munc13-1 intensity shows a significant positive correlation with *N/AZ* (middle panel) and the lack of correlation with *Pv* (right panel). r$_s$: Spearman's rank correlation coefficient s.o. stratum oriens, s.p. stratum pyramidale, s.r. stratum radiatum, s.lm. stratum lacunosum-moleculare.

The online version of this article includes the following source data for figure 3:

**Source data 1.** STED analysis of Munc13-1 and PSD-95 immunosignals at functionally characterized PC–FSIN synapses.

## Variable size and molecular content of Munc13-1 clusters in glutamatergic AZs on Kv3.1b + INs as revealed by SDS-FRL

To investigate the relationship between the size of AZs and the amounts of Munc13-1, we obtained replicas from the CA1 region of age-matched mouse hippocampus. First, we verified the specificity of our labeling using two Munc13-1 antibodies recognizing non-overlapping epitopes (*Figure 4—figure supplement 1*). We then performed double immunogold labeling for Kv3.1b and Munc13-1 (*Figure 4*). We used the Kv3.1b potassium channel subunit to identify fractured membrane segments of parvalbumin-positive FSINs (*Weiser et al., 1995*). AZs fractured on these Kv3.1b + IN somata and proximal dendrites are highly variable in size (mean = 0.071 ± 0.014 µm$^2$, CV = 0.43 ± 0.06, n = 4 reactions in three mice) and contain variable number of gold particles labeling Munc13-1 (mean = 26.0 ± 5.1 gold, CV = 0.49 ± 0.08, n = 4; *Figure 4C–G*). Visual inspection of the EM images revealed that large AZs had many gold particles and small ones had fewer. Indeed, a significant positive correlation was observed between the AZ size and the Munc13-1 gold number in four experiments of three mice (*Figure 4I*). If Munc13-1 had a tight correlation with the AZ area, then its density should be uniform and synapse size independent. Plots showing the Munc13-1 density vs. the AZ area revealed substantial (mean CV = 0.33 ± 0.07) and slightly synapse size-dependent variability (*Figure 4J*). Synapses with identical size could have a 10-fold difference in their Munc13-1 content, suggesting large variability in either the RS density or the amounts of Munc13-1 per RS. To exclude the possibility that a significant source of this variability is technical, we carried out PSD-95 labeling experiments (*Figure 5*). The number of gold particles for PSD-95 showed an extremely tight, positive correlation with the synapse area (r$_s$ = 0.96), resulting in a size-independent uniform PSD-95 density (*Figure 5C and D*). The exceptionally small variability in the PSD-95 density (CV = 0.09) demonstrates the capability of SDS-FRL method to reveal uniform densities of synaptic molecules with a small variance. Because the variability in Munc13-1 density is substantially higher (CV = 0.33 ± 0.07) with similar mean values (Munc13-1: 383 ± 71 gold/µm$^2$ vs PSD-95 in dendrites: 497 ± 45 gold/µm$^2$) and because we demonstrate that this cannot be the consequence of a lower labeling efficiency of our Munc13-1 Ab (see Materials and methods), we concluded that the observed synapse to synapse variation in the Munc13-1 density must have a biological origin.

Next, we investigated the sub-synaptic distribution of Munc13-1 as it has been suggested to have a clustered distribution in AZs (*Rebola et al., 2019*; *Sakamoto et al., 2018*; *Tang et al., 2016*) and the clusters represent the RSs. First, we measured mean nearest-neighbor distances (NND) between gold particles in the AZs and compared them to random particle distributions. The mean NND distances were significantly smaller than those of randomly distributed gold particles (data: 0.026 ± 0.01 µm, random: 0.033 ± 0.009 µm, n = 159, p<0.001, Wilcoxon signed-rank test; *Figure 4K*). A previous study from our laboratory demonstrated that Ripley's H-function analysis could reveal clustered distribution of synaptic molecules, including Munc13-1 in cerebellar synapses

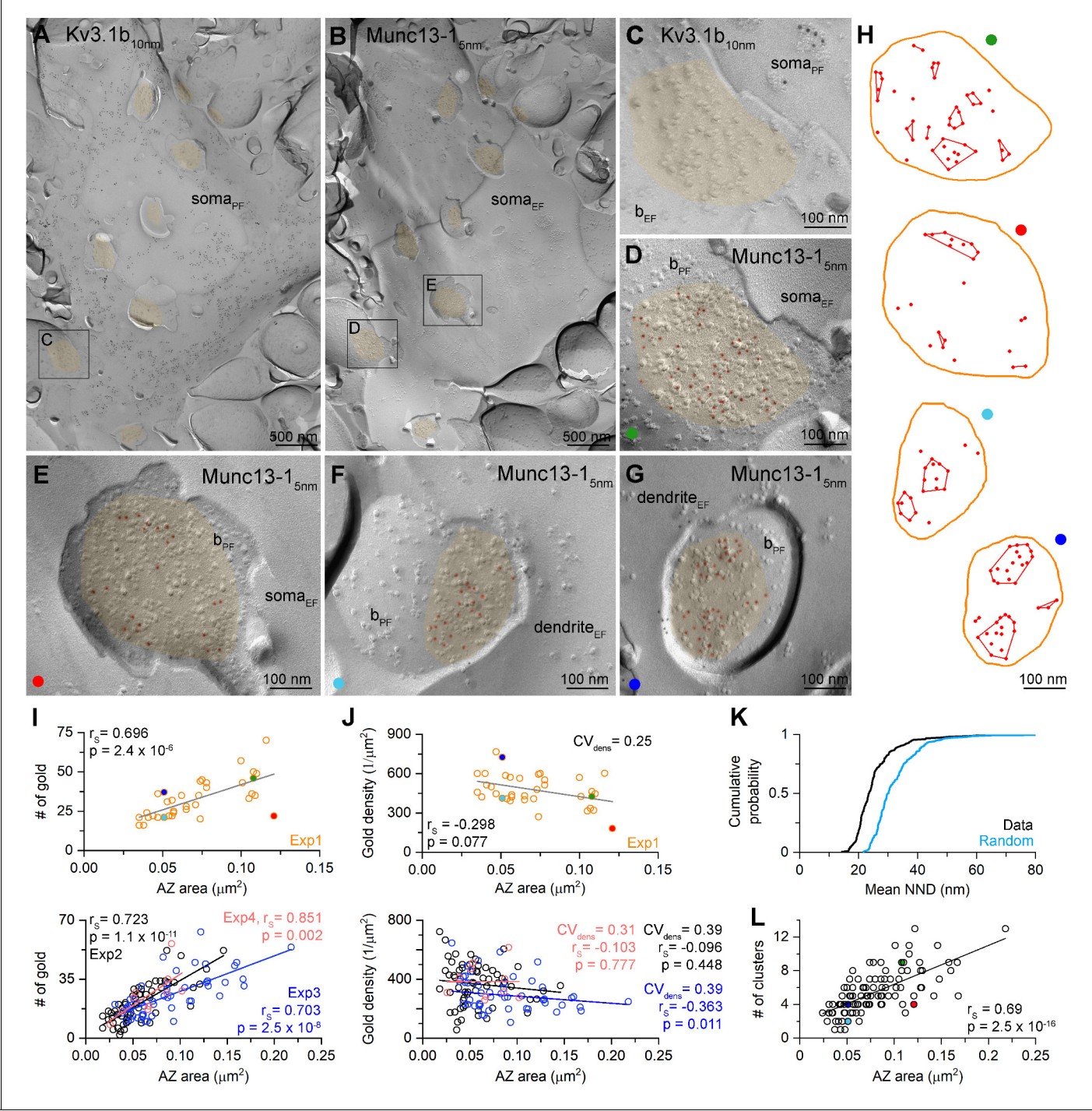

**Figure 4.** Density of Munc13-1 shows large variability in AZs targeting Kv3.1b + cells in hippocampal CA1 area as revealed by SDS-FRL. (**A, B**) Low-magnification EM images of corresponding protoplasmic-face (soma$_{PF}$, **A**) and exoplasmic-face (soma$_{EF}$, **B**) membranes of a Kv3.1b + cell body in the stratum oriens. AZs fractured onto the somatic plasma membranes are highlighted in orange. (**C, D**) High-magnification images of the boxed areas from (**A**) and (**B**) show matching EF and PF membranes of a bouton (b$_{EF}$ and b$_{PF}$) attached to the Kv3.1+ cell. 5 nm gold particles (highlighted in red) labeling Munc13-1 are accumulated in the AZ (orange) of the bouton. (**E–G**) Other examples of Munc13-1 labeled AZs attached to Kv3.1b + somata or dendrites. (**H**) Distribution and cluster identification of gold particles labeling Munc13-1 in the AZs shown in (**D–G**) by DBSCAN analysis (epsilon = 31 nm, minimum number of particles per cluster = 2). (**I**) Number of Munc13-1 gold particles as a function of AZ area. Data from Exp1 (n = 36) is shown on the upper panel, additional three experiments are shown on the lower panel (Exp2, n = 65; Exp3, n = 48, Exp4 = 10 from three mice). The four AZs shown in (**D–G**) are indicated by their corresponding colors. (**J**) Density of Munc13-1 gold particles as a function of AZ area. Data from Exp1 (n = 36) is shown on the upper panel, additional three experiments are shown on the lower panel. (**K**) Cumulative distribution of mean NNDs (per AZ) of Munc13-1

*Figure 4 continued on next page*

*Figure 4 continued*

gold particles (n = 159 AZs) and mean NNDs of randomly distributed particles within the same AZs (generated from 200 random distributions per AZ, p<0.001, Wilcoxon test). (**L**) Number of Munc13-1 gold particle clusters (estimated by DBSCAN analysis, n = 105 AZs) as a function of AZ area. Colored symbols represent the AZs shown in panels (**D–G**). $r_S$, Spearman's rank correlation coefficient.

The online version of this article includes the following source data and figure supplement(s) for figure 4:

**Source data 1.** Density of Munc13-1 shows large variability in AZs targeting Kv3.1b + cells in hippocampal CA1 area as revealed by SDS-FRL.

**Figure supplement 1.** Specificity test of the Munc13-1 immunolabeling.

(*Rebola et al., 2019*). We performed this analysis on 159 AZs and found that in 66% of the AZs the distribution of gold particles was compatible with clustering (p<0.05, MAD test). We then used DBSCAN (*Ester et al., 1996*) to identify the Munc13-1 clusters in these 105 AZs. DBSCAN requires two user-defined parameters: ε (nm), which is the maximum distance between two localization points to be assigned to the same cluster, and *MinPts*, the minimum number of points within a single cluster. We systematically changed the ε value from 1 to 100 nm and found the largest difference between the data and the random distributions at ε = 31 nm. We then determined the mean number of clusters ($N_c$ = 5.4 ± 2.5) with this ε value and a *MinPts* of 2. We then tested the effects of changing ε and *MinPts* on $N_c$ (ε = 21, *MinPts* = 2, $N_c$ = 5.7 ± 2.7; ε = 41, *MinPts* = 2, $N_c$ = 4.0 ± 1.8; ε = 31, *MinPts* = 3, $N_c$ = 3.8 ± 1.8) and found that changing these parameters within plausible values results in a moderate change in $N_c$. The average of ~5 clusters per AZ is remarkably similar to the N/AZ (4.9 ± 3.7), supporting the notion that Munc13-1 clusters are indeed the molecular equivalents of the functional RSs (*Sakamoto et al., 2018*). When the number of clusters were plotted against the AZ area, a significant positive correlation was found (*Figure 4L*). However, the number of clusters also varied fourfold in synapses of identical sizes, resulting in a CV of 0.36 in the cluster density (mean:

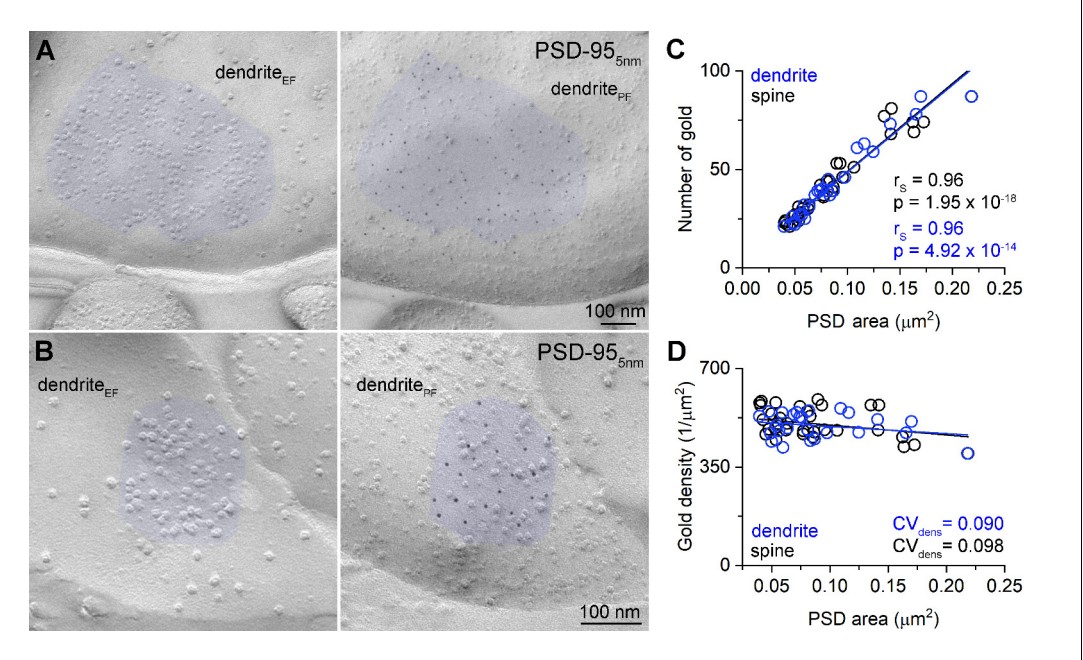

**Figure 5.** Uniform PSD-95 immunolabeling density in the PSDs. (**A**, **B**) Two mirror replica pairs showing excitatory postsynaptic densities (PSDs) on dendrites in the CA1 area. PSD area is identified by the accumulation of intramembrane particles on the exoplasmic-face dendritic membranes (dendrite_EF) highlighted by blue (left). The corresponding protoplasmic-face of the same dendrite (dendrite_PF) is labeled for PSD-95 with 5 nm gold particles (right). (**C**) Number of gold particles labeling PSD-95 as a function of PSD area in dendritic shaft (n = 25) and dendritic spine (n = 32) synapses. (**D**) Density of gold particles labeling PSD-95 as a function of PSD area in dendritic shafts (n = 25, $r_s$ = −0.115, p=0.583) and dendritic spines (n = 32, $r_s$ = −0.326, p=0.069). $r_s$, Spearman's rank correlation coefficient.

The online version of this article includes the following source data for figure 5:

**Source data 1.** Uniform PSD-95 immunolabeling density in the PSDs.

73 ± 27 clusters/μm$^2$ AZ area, n = 105). We also noticed that not only the cluster density varies, but the Munc13-1 content of the clusters (4.5 ± 3.0 gold/cluster, CV = 0.67, n = 571) is also highly variable (for individual AZs see *Figure 4H*). Finally, we measured the nearest-neighbor inter-cluster distance and obtained a mean of 85 ± 34 nm (n = 568 distances from 102 AZs) that is consistent with the spacing and size of docked synaptic vesicles.

## Quantitative STED analysis reveals highly variable amounts of Munc13-1 in excitatory synapses of identical sizes

Our replica-labeling experiments reveal large variability in the Munc13-1 content of synapses with identical sizes, which is the consequence of both the variability in the cluster density and the molecular content of the clusters. We believe that the replica-labeling is the most appropriate method for quantitative analysis of sub-synaptic distributions of molecules due to its high resolution and sensitivity, but unfortunately, it is impossible to perform SDS-FRL in synapses that had been functionally characterized due to the random fracturing of the tissue. Because of this limitation, we developed the above described postembedding, multiplexed immunofluorescent reaction with which we could molecularly characterize functionally tested individual synapses (*Holderith et al., 2020*). In our final set of experiments, we aimed to compare the results of the postembedding reactions to those obtained with SDS-FRL.

We randomly selected and serially sectioned proximal dendritic segments of two in vitro recorded FSINs (*Figure 6*). The sections were then immunoreacted for Munc13-1 and PSD-95 in consecutive labeling rounds, and their reaction strengths were quantitatively analyzed on the STED images. First, we performed the analysis on 200 nm thick sections (the usual section thickness in our protocol) and focused on en face synapses where the pre- and postsynaptic specializations are present in a single section and therefore no 3D reconstruction is needed from serial sections (*Figure 6C*). In the two examined cells relative Munc13-1 and PSD-95 intensities showed a loose correlation (*Figure 6D*). More importantly, the PSD-95 normalized Munc13-1 labeling showed a substantial variability (Cell 1: CV = 0.42; Cell 2: CV = 0.40) and a slight synapse size- (PSD-95 intensity) dependence, like that obtained with SDS-FRL (compare *Figure 4J* with *Figure 6E*). Because the orientation of the functionally characterized synapses related to the sectioning plane is random, i.e. is not always perpendicular or vertical, we repeated these experiments using 70 nm section thickness and performed full 3D reconstruction of the synapses from serial sections (*Figure 6—figure supplement 1*). As can be seen in the superimposed STED images in *Figure 6—figure supplement 1C*, the relative proportion of cyan (Munc13-1) and red (PSD-95) signals varies substantially, resulting in a large variability in the PSD-95 normalized Munc13-1 signal (CV = 0.40, *Figure 6—figure supplement 1D,E*) again consistent with our SDS-FRL results and indicating that the different amounts of Munc13-1 in synapses with identical N/AZ are likely to be of biological origin.

## Discussion

Data obtained in three independent series of experiments indicate a substantial variability in the molecular content of presynaptic RSs within individual AZs. (1) By determining the N with quantal analysis and subsequently the amounts of Munc13-1 molecules in the functionally characterized AZs, we revealed that AZs with similar Ns have very different amounts of Munc13-1 (*Figure 3J*). (2) When populations of synapses on FSINs were examined with multiplexed postembedding immunolabeling and STED analysis, the PSD-95 (synapse size)-normalized Munc13-1 immunolabeling showed large variability (*Figure 6E*). (3) Finally, SDS-FRL, the currently known most sensitive and highest resolution immunolocalization method, demonstrated large variability in the Munc13-1 density in AZs on FSINs and subsequently revealed a synapse size-independent variability in the number of Munc13-1 clusters and in the Munc13-1 content of such clusters (*Figure 4J,L*).

It is well known that synapses made by molecularly identical presynaptic nerve cells on molecularly identical postsynaptic cells can show large structural and functional variability (reviewed by *Pulido and Marty, 2017*). In the present study, we examined the connections between hippocampal CA1 PCs and FSINs in adult mice and revealed large variability in uEPSC amplitudes (from 3 to 500 pA, CV = 1) evoked by single PC APs. This large amplitude variability is also present in dendritically unfiltered EPSCs and for both morphologically defined basket and bistratified cells. Quantal analysis demonstrated that variability in N has the largest contribution to the variance in uEPSC amplitudes,

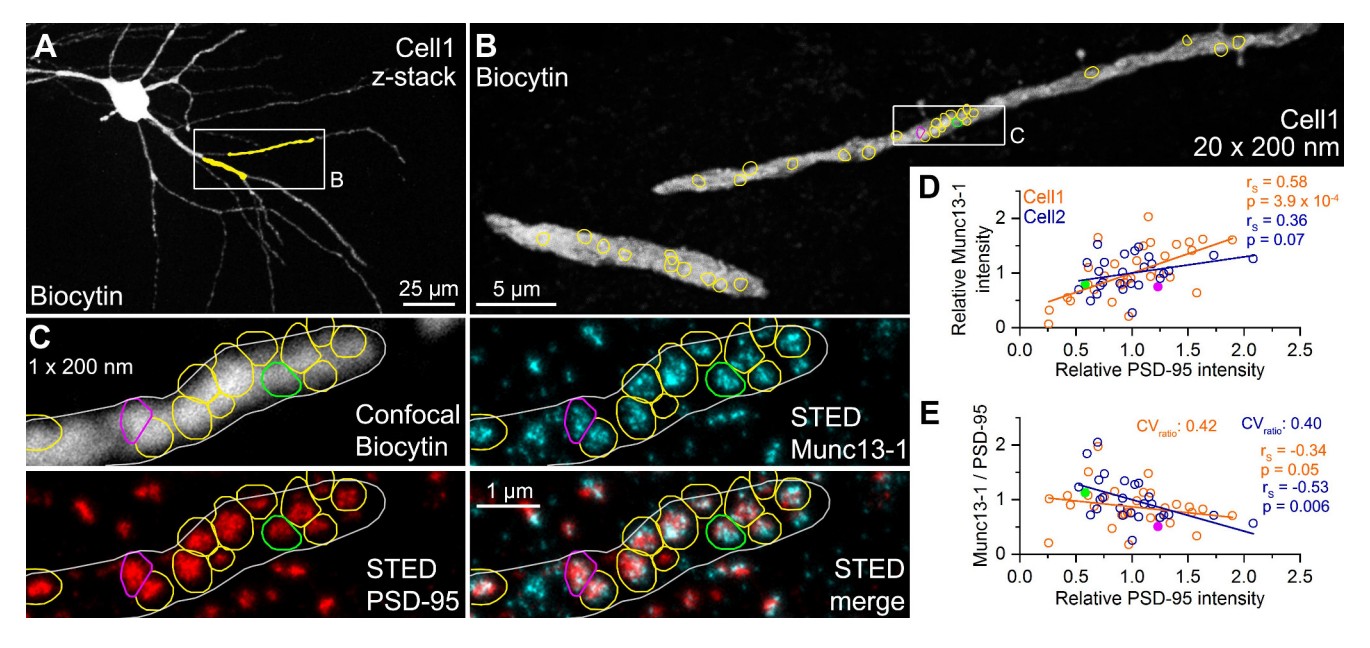

**Figure 6.** Quantitative STED analysis reveals highly variable Munc13-1 signal in excitatory synapses on FSIN dendrites. (**A**) Confocal maximum intensity projection image of a biocytin-filled FSIN (Cell1, soma, and basal dendrites in the str. oriens are shown). The dendritic segments that were re-sectioned and analyzed are highlighted in yellow. (**B**) Reconstruction of the re-sectioned dendritic segments (from twenty 200 nm thick sections) shown in (**A**). Colored circles indicate en face excitatory synapses (n = 33) identified by Munc13-1 and PSD-95 double immunolabeling. (**C**) STED analysis of Munc13-1 and PSD-95 immunofluorescent signals on a single 200 nm thick section (shown in the boxed area in (**B**)). The biocytin-filled dendrite shown in a confocal image (top left) is outlined by a white line. Colored regions of interests (ROIs) represent en face synapses based on the Munc13-1 and PSD-95 immunosignals. (**D**) Relative Munc13-1 intensity as a function of relative PSD-95 signal in individual synapses. Symbols represent mean normalized integrated fluorescent intensities in individual synapses (Cell1, n = 33; Cell2, n = 26). Magenta and green filled symbols indicate the corresponding color-coded synapses shown in (**C**). Note that the two synapses have very similar Munc13-1 content although their PSD-95 reactivity is ~2.5-fold different. (**E**) Munc13-1 to PSD-95 ratio as a function of relative PSD-95 intensity in individual synapses. Magenta and green filled symbols indicate the corresponding color-coded synapses shown in (**C**). $r_s$, Spearman's rank correlation coefficient.

The online version of this article includes the following source data and figure supplement(s) for figure 6:

**Source data 1.** Quantitative STED analysis reveals highly variable Munc13-1 signal in excitatory synapses on FSIN dendrites.

**Figure supplement 1.** Quantitative STED analysis on 70 nm thick sections reveals high variance in Munc13-1 signal in excitatory synapses on FSIN dendrites.

**Figure supplement 1—source data 1.** Quantitative STED analysis on 70 nm thick sections reveals high variance in Munc13-1 signal in excitatory synapses on FSIN dendrites.

which is the consequence of an approximately equal variability in the number of synapses per connection (2.3 ± 1.6, CV = 0.68, from 1 to 7) and the *N/AZ* (4.9 ± 3.7, CV = 0.75, from 1 to 17). PC to FS basket cell synaptic connections are mediated by a remarkably similar number of synapses in human neocortex (mean = 3.3, range: 1–6; *Molnár et al., 2016*), cat visual cortex (mean = 3.4, range: 1–7; *Buhl et al., 1997*), rat neocortex (mean = 2.9, range 1–6; *Molnár et al., 2016*), and mouse hippocampus (mean = 2.3, range: 1–7; present study). It seems that it is not a unique feature of PC–FSIN connections because a very similar number (mean = 2.8, range: 1–6) was found when CA1 PCs to oriens-lacunosum-moleculare (O-LM) IN connections were examined in juvenile rats (*Biró et al., 2005*). All data taken together demonstrate that the multi-synapse connection between PCs and GABAergic local circuit INs is an evolutionary conserved feature of cortical networks. As mentioned above, currently it is unknown why PC to IN connections are mediated by multiple (mean of ~3) and variable number (1 – 7) of synaptic contacts.

Unlike the number of synapses per connection, when the *N/AZ* was compared, a much larger variability and a species-specific difference was found. *Molnár et al., 2016* reported that the *N/AZ* was ~4 times larger in human (~6) compared to rat (1.6) cortical PC–FSIN connections. It is 4.9 for the same connection in adult mouse hippocampus, which is very similar to that found in mouse

cultured hippocampal neurons (4.9 in *Sakamoto et al., 2018* and 4.2 in *Ariel et al., 2012*; but 8.7 in *Maschi and Klyachko, 2017*). The difference in *N*/AZ between human and rat was accompanied by a larger AZ size in human (0.077 $\mu m^2$), which is again similar to that obtained in our present study in adult mice (0.071 $\mu m^2$), indicating that both in human and mice a RS occupies (or needs) approximately the same AZ area. The positive correlation between the docked vesicles and the AZ area (*Molnár et al., 2016*; *Schikorski and Stevens, 1999*) is consistent with a model in which the *N* scales linearly with the AZ area and each independent RS is built up from the same number of molecules (*Sakamoto et al., 2018*). However, when not only the mean, but the variance in the available data is also considered, a more complex picture emerges. First, there is large variability in the number of docked vesicles in AZs with identical sizes (Figure 3—figure supplement 1 in *Molnár et al., 2016*), which might reflect variability in RS density, but an incomplete docking site/RS occupancy cannot be excluded. Such incomplete RS occupancy cannot explain our data showing that AZs with the same amount of PSD-95 (same size) have >3 fold variability in *N*. Thus, it seems that variability in the docking site occupancy might not be the main source of variability, but the actual RS density seem to be variable. A similar large variability is present in the data of Figure 3c in *Sakamoto et al., 2018* in the correlation between the readily releasable pool size ($N_{RRP}$) and the number of labeled Munc13-1 molecules. The almost identical number of RSs and $N_{RRP}$ indicates a docking site occupancy close to one in their cultured hippocampal neurons again arguing for the variability in either the RS density or in the number of Munc13-1 molecules per RS. Our high-resolution SDS-FRL experiments provide direct evidence for both: substantial variability in the Munc13-1 cluster (i.e. RS) density in AZs and in the number of Munc13-1 molecules per cluster (*Figure 4H,L*).

One consequence of the variable number of docked vesicles or RS density is that the inter RS distance varies substantially in AZs of identical sizes. One possible consequence of that is that the RSs might not function independently when they are close enough to 'see' substantial amounts of $Ca^{2+}$ from the neighboring RSs. Our data, showing that the average *Pv* of the RSs does not depend on the *N*/AZ (*Figure 2J*), together with that of *Sakamoto et al., 2018*, demonstrating that *Pv* does not depend on the $N_{RRP}$, strongly indicate that the average *Pv* does not depend on the size of AZ. A pervious study from our laboratory (*Holderith et al., 2012*) described that the probability with which release occurs at hippocampal synapses ($P_R$) depends on the AZ size. We would like to stress that this probability ($P_R$) is the function of both the *Pv* and N [$P_R$ = 1-(1-Pv)$^N$]; therefore, the synapse size-dependent increase in *N* fully explains our previous and current results.

What might be the consequence of the variable amounts of Munc13-1 in RSs? Munc13-1 is an evolutionarily conserved presynaptic protein that is essential for docking and priming vesicles for release (*Augustin et al., 1999*; *Betz et al., 2001*; *Brockmann et al., 2020*; *Imig et al., 2014*; *Jahn and Fasshauer, 2012*; *Ma et al., 2011*; *Varoqueaux et al., 2002*); therefore, it can be hypothesized that the amount of this molecule might have an effect on the docking site occupancy or the priming state of the vesicles. MPFA only allows the determination of *Pv*, a probability that depends on the probability of the RS being occupied ($P_{occ}$) and on the probability of a docked vesicle being released ($P_{succ}$; *Neher, 2017*). Two lines of evidence indicate that $P_{occ}$ is high at neocortical/hippocampal glutamatergic synapses. As mentioned above, *Sakamoto et al., 2018* came to this conclusion from the similar $N_{RRP}$ and *Molnár et al., 2016* examined the number of docked vesicles at cortical PC–FSIN synapses and determined *N*, and found rather similar values for both human and rat synapses, arguing for a $P_{occ}$ of ~0.8 that is similar to that found at the Calyx of Held (*Neher, 2010*), but larger than at cerebellar IN synapses (*Pulido et al., 2015*). Thus, it seems that variability in $P_{occ}$ might not be the major consequence of the variable amounts of Munc13-1 per RS, indicating that the priming state of the vesicles and therefore the $P_{succ}$ might be affected. Heterogeneity in the *Pv* for different vesicles has been demonstrated at the AZs of the Calyx of Held (reviewed by *Neher, 2017*). Here, approximately half of the vesicles have high and the other halves have low *Pv*. Furthermore, there is also data indicating further heterogeneity in the *Pv* of the fast releasing (high *Pv*) vesicles in the Calyx (normally primed and superprimed vesicles; *Taschenberger et al., 2016* and in hippocampal synapses as well *Hanse and Gustafsson, 2001*; *Schlüter et al., 2006*). Whether such high- and low-*Pv* vesicles are intermingled within individual AZs or are segregated to distinct AZs is unknown. It is just as unknown whether the normally and superprimed vesicles need different amounts of Munc13-1 or not. It is noteworthy that the priming efficacy of Munc13-1 depends on its interaction with RIM and RIM binding protein (*Brockmann et al., 2020*) therefore predicting the functional consequence of the different amounts of Munc13-1 per RS

might require the determination of these molecules in individual Munc13-1 clusters. A recent study using superresolution imaging of vesicle release from cultured hippocampal neurons provided strong evidence for the heterogeneity in $Pv$ among RSs within individual AZs. *Maschi and Klyachko, 2020* demonstrated that the $Pv$ of centrally located RSs is higher and participate more frequently in multivesicular release (MVR) than those that are located at the periphery of the AZs. These data taken together indicate substantial variability in $Pv$ among RSs, which is more likely to be the consequence of variable $P_{succ}$, the relationship of which to the amounts of Munc13-1 molecules remains to be seen.

A recent study applied superresolution light microscopy to examine the spatial relationship between pre- and postsynaptic molecules and came to the conclusion that key molecules are arranged in transsynaptic nanocolumns (*Tang et al., 2016*). Specifically, the number of RIM1 clusters in the AZ was similar to that of PSD-95 in the PSD and they face one another in a remarkable fashion. Other presynaptic molecules such as Munc13-1 and bassoon were also clustered, but their numbers were higher and lower, respectively than that of PSD-95, slightly complicating the picture. Our present data and that of *Rebola et al., 2019* showing clustering of Munc13-1 in the AZs is consistent with such a model. However, our data demonstrating virtually zero synapse size-independent variability in the amounts of PSD-95 molecules, but large variability in the Munc13-1 seems to be inconsistent with such a model. In addition, the distribution of our PSD-95 labeling does not indicate strong intra synapse clustering either. A more direct evidence against the nanocolumn organization as a universal feature of all glutamatergic synapses comes from a previous study form our laboratory, demonstrating that postsynaptic AMPA receptors do not show clustered sub-synaptic distribution in cerebellar and hippocampal glutamatergic synapses on GABAergic INs (*Szoboszlay et al., 2017*). All these data taken together indicates that the nanocolumn arrangement of pre- and postsynaptic key signaling molecules is not a universal feature of all glutamatergic synapses.

Our results are also compatible with the concept that individual cortical synapses release more than a single vesicle from an AZ upon the arrival of a single AP (called MVR; *Biró et al., 2006*; *Christie, 2006*; *Maschi and Klyachko, 2020*; *Pulido et al., 2015*; *Rudolph et al., 2015*; *Wadiche and Jahr, 2001*). The occurrence of MVR is the function of $N/AZ$ and $Pv$. All available data indicate that cortical/hippocampal excitatory and inhibitory synaptic AZs contain multiple RSs, the number of which positively correlates with the size of the AZ, fulfilling one essential requirement of MVR. The average $Pv$, however, is much more heterogeneous. The most compelling evidence for variable $Pv$ in distinct boutons is the postsynaptic target cell type-dependent variability in $Pv$ and short-term plasticity (*Éltes et al., 2017*; *Koester and Johnston, 2005*; *Losonczy et al., 2002*; *Pouille and Scanziani, 2004*; *Reyes et al., 1998*; *Rozov et al., 2001*; *Scanziani et al., 1998*; *Thomson, 1997*). A previous study from our laboratory demonstrated that $Pv$ at hippocampal CA1 PC to O-LM cell synapses is so low that the occurrence of MVR is negligible under physiological conditions (*Biró et al., 2005*). However, the $Pv$ at PC–FSIN synapses is an order of magnitude higher (~0.4) than that at PC–O-LM synapses and given an average of five RSs per AZ, the probability of MVR is around 70%. We would like to emphasize that $Pv$ at CA3 to CA1 PC synapses is probably in between these values, indicating that the occurrence of MVR is much less prominent. The degree of postsynaptic receptor occupancy is a key issue when the functional consequence of MVR is considered. If the occupancy is high (e.g. cerebellar climbing fiber to Purkinje cell synapses; *Harrison and Jahr, 2003* or at cerebellar molecular layer IN synapses, *Auger et al., 1998*; *Nusser et al., 1997*), the effect of simultaneously released multiple vesicles is small (not necessarily zero, because of the effect on the time course of the postsynaptic response; *Rudolph et al., 2011*). However, increasing evidence indicates that receptor occupancy is relatively low at most central glutamatergic synapses, allowing the postsynaptic cell to detect the number of simultaneously released vesicles within a single synapse either linearly or sublinearly. Our result, showing no correlation between the $q$ and $N/AZ$ (*Figure 2J*), is also consistent with this. Such MVR operation of synapses by increasing the reliability of transmission and reducing stochastic trial-to-trial variability might provide an important circuit element with which perisomatic GABAergic inhibition is recruited reliably by an active ensemble of PCs.

# Materials and methods

### Key resources table

| Reagent type (species) or resource | Designation | Source or reference | Identifiers | Additional information |
|---|---|---|---|---|
| Antibody | (Rabbit polyclonal) anti-Munc13-1 | Synaptic systems | Cat#126–103; RRID:AB_887733 | (1:200) |
| Antibody | (Guinea pig polyclonal) anti-Munc13-1 | Custom made by Synaptic systems | | (1:200) |
| Antibody | (Guinea pig polyclonal) anti-panAMPAR | Frontiers | Cat#Af580; RRID:AB_257161 | (1:200) |
| Antibody | (Guinea pig polyclonal) anti-PSD95 | Synaptic systems | Cat#124–014; RRID:AB_2619800 | (1:200) (1:500 for FRL) |
| Antibody | (Rabbit polyclonal) anti-vGluT1 | Synaptic systems | Cat#135–302; RRID:AB_887877 | (1:200) |
| Antibody | (Donkey anti-rabbit polyclonal) Alexa488 | Jackson | Cat# 711-545-152, RRID:AB_2313584 | (1:200) |
| Antibody | (Donkey anti-Guinea pig polyclonal) Alexa488 | Jackson | Cat# 706-545-148, RRID:AB_2340472 | (1:200) |
| Antibody | (Goat anti-Guinea pig polyclonal) Abberior STAR 635P | Abberior | Cat#2-0112-007-1 | (1:200) |
| Antibody | (Goat anti-rabbit polyclonal) Abberior STAR 635P | Abberior | Cat#2-0012-007-2 | (1:200) |
| Antibody | (Guinea pig polyclonal) anti-Cav2.1 | Synaptic systems | Cat#152 205; RRID:AB_2619842 | (1:3000) |
| Antibody | (Rabbit polyclonal) anti-Kv3.1b | Synaptic systems | Cat#242 003; RRID:AB_11043175 | (1:1600) |
| Antibody | (Goat anti-rabbit polyclonal) 5 nm gold conjugated | British Biocell International | EM.GAR5 | (1:80) |
| Antibody | (Goat anti- guinea pig polyclonal) 5 nm gold conjugated | British Biocell International | EM.GAG5 | (1:80) |
| Antibody | (Goat anti-rabbit polyclonal) 10 nm gold conjugated | British Biocell International | EM.GAR10 | (1:80, 1:100) |
| Antibody | (Donkey anti-guinea pig polyclonal) 12 nm gold conjugated | Jackson ImmunoResearch | 706-205-148 RRID:AB_2340465 | (1:25) |
| Antibody | Streptavidin Cy3 coupled | Jackson | Cat# 016-160-084, RRID:AB_2337244 | (1:100) |
| Chemical compound, drug | Sodium dodecyl sulfate | Sigma | 71725–100G | |
| Chemical compound, drug | Bovine Serum Albumin | Sigma | A2153-50G | |
| Chemical compound, drug | Paraformaldehyde | Molar Chemicals | Cat#BC0487491 | |

*Continued on next page*

*Continued*

| Reagent type (species) or resource | Designation | Source or reference | Identifiers | Additional information |
|---|---|---|---|---|
| Chemical compound, drug | Uranyle acetate | TAAB | Cat#U008 | |
| Chemical compound, drug | Durcupane ACM Resin Single component A | Sigma-Aldrich | Cat#44611 | |
| Chemical compound, drug | Durcupane ACM Resin Single component B | Sigma-Aldrich | Cat#44612 | |
| Chemical compound, drug | Durcupane ACM Resin Single component C | Sigma-Aldrich | Cat#44613 | |
| Chemical compound, drug | Durcupane ACM Resin Single component D | Sigma-Aldrich | Cat#44614 | |
| Chemical compound, drug | Picric acid | Sigma-Aldrich | Cat#197378 | |
| Chemical compound, drug | Triton X100 | VWR Chemicals | Cat#9002-93-1 | |
| Chemical compound, drug | Slowfade Diamond | Invitrogen | Cat#S36967 | |
| Chemical compound, drug | Tris Base | Sigma-Aldrich | Cat#252859 | |
| Chemical compound, drug | Tris–HCl | Sigma-Aldrich | Cat#T3253 | |
| Chemical compound, drug | $NaH_2PO_4$ | Sigma-Aldrich | Cat#S0751 | |
| Chemical compound, drug | $Na_2HPO_4$ | Sigma-Aldrich | Cat#S9763 | |
| Chemical compound, drug | BlottoA | Santa Cruz Biotechnology | Cat#Sc2333 | |
| Chemical compound, drug | Normal goat serum (NGS) | Vector Laboratories | Cat#S-1000 | |
| Chemical compound, drug | Bovine serum albumin (BSA) | Sigma-Aldrich | Cat#A2153 | |
| Chemical compound, drug | Ketamine | Produlab Pharma B.V. | #2302/2/07, 10% | |
| Chemical compound, drug | Xylasine | Produlab Pharma B.V. | #2303/3/07, 20 mg / ml | |
| Chemical compound, drug | Pipolphene | EGIS Gyógyszergyár Zrt. | #OGYI-T-3086/01, 25 mg / ml | |

*Continued on next page*

*Continued*

| Reagent type (species) or resource | Designation | Source or reference | Identifiers | Additional information |
|---|---|---|---|---|
| Chemical compound, drug | Sucrose | Sigma-Aldrich | Cat#S5016 | |
| Chemical compound, drug | KCl | Sigma-Aldrich | Cat#P3911 | |
| Chemical compound, drug | $NaHCO_3$ | Sigma-Aldrich | Cat#S6014 | |
| Chemical compound, drug | $CaCl_2$ | Sigma-Aldrich | Cat#C5080 | |
| Chemical compound, drug | $MgCl_2$ | Sigma-Aldrich | Cat#M2670 | |
| Chemical compound, drug | $NaH_2PO_4$ | Sigma-Aldrich | Cat#S0751 | |
| Chemical compound, drug | glucose | Sigma-Aldrich | Cat#G7528 | |
| Chemical compound, drug | NaCl | Sigma-Aldrich | Cat#S9888 | |
| Chemical compound, drug | K-gluconate | Sigma-Aldrich | Cat#P1847 | |
| Chemical compound, drug | Cesium methanesulfonate | Sigma-Aldrich | Cat#C1426 | |
| Chemical compound, drug | Creatinine phosphate | Sigma-Aldrich | Cat#27920 | |
| Chemical compound, drug | HEPES | Sigma-Aldrich | Cat#H7523 | |
| Chemical compound, drug | ATP disodium salt | Sigma-Aldrich | Cat#A2383 | |
| Chemical compound, drug | GTP sodium salt | Sigma-Aldrich | Cat#G8877 | |
| Chemical compound, drug | Biocytin | Sigma-Aldrich | Cat#B4261 | |
| Strain, strain background (include species and sex here) | Mouse, male, female C57Bl6/J | Jackson | Cat# JAX:000664, RRID:IMSR_JAX:000664 | |
| Strain, strain background (include species and sex here) | Mouse, male, female Tg(Chrna2-Cre)OE25Gsat/Mmucd | Jackson | RRID:MMRRC_036502-UCD | |
| Software, algorithm | Image J | National Institute of Health | https://imagej.nih.gov/ij; RRID:SCR_003070 | |

*Continued on next page*

*Continued*

| Reagent type (species) or resource | Designation | Source or reference | Identifiers | Additional information |
|---|---|---|---|---|
| Software, algorithm | Hyperstack stitcher (ImageJ plugin) | This paper, 3D Histech | http://www.nusserlab.hu/ | |
| Software, algorithm | Adobe Photoshop CS3 | Adobe | https://www.adobe.com/hu/products/photoshop.html | |
| Software, algorithm | Origin 2018 | OriginLab | https://www.originlab.com/ | |
| Software, algorithm | Multiclamp (version 2.1) | Axon Instruments/ Molecular Devices | https://www.moleculardevices.com/ | |
| Software, algorithm | Clampex (version 10.3) | Axon Instruments/ Molecular Devices | https://www.moleculardevices.com/ | |
| Software, algorithm | GoldExt | Nusser Lab | http://www.nusserlab.hu/ | |
| Software, algorithm | Statistica | TIBCO Software Inc. | https://www.tibco.com | |
| Other | Vibratome VT1200S | Leica | https://www.leica-microsystems.com/ | |
| Other | Ultramicrotome EM UCT | Leica | https://www.leica-microsystems.com/ | |
| Other | Abberior Instruments Expert Line STED microscope | Abberior Instruments | https://www.abberior.com/ | |
| Other | Olympus FV1000 Confocal microscope | Olympus | https://www.olympus-lifescience.com/ | |
| Other | Multiclamp 700B amplifier | Axon Instruments/Molecular Devices | https://www.moleculardevices.com/ | |
| Other | DMZ Zeits Puller | Zeitz | https://www.zeitz-puller.com/ | |
| Other | Borosilicate glass capillary | Sutter Instruments | Cat# BF150-86-10 | |
| Other | Superfrost Ultra plus slide | Thermoscientific | http://www.thermoscientific.com | |
| Other | PapPen | ThermoFisher Scientific | Cat# 008899 | |
| Other | Olympus BX51 microscope | Olympus | https://www.olympus-lifescience.com/ | |
| Other | Nikon Eclipse FN1 microscope | Nikon | https://www.nikon.com/ | |
| Other | Leica EM ACE900 Freeze Fracture System | Leica Microsystems | https://www.leica-microsystems.com/ | |
| Other | Leica HPM100 High Pressure Freezing System | Leica Microsystems | https://www.leica-microsystems.com/ | |
| Other | Jeol JEM1011 Transmission electronmicroscope | Jeol | https://www.jeol.co.jp/ | |

## Animals

Animals were housed in the vivarium of the Institute of Experimental Medicine in a normal 12 hr/12 hr light/dark cycle and had access to water and food ad libitum. All the experiments were carried out according to the regulations of the Hungarian Act of Animal Care and Experimentation 40/2013 (II.14) and were reviewed and approved by the Animal Committee of the Institute of Experimental Medicine, Budapest.

## SDS-digested freeze-fracture replica-labeling

Three C57Bl/6J (P49–P63) male and a P49 female mice were deeply anesthetized and were transcardially perfused with ice-cold fixative containing 2% formaldehyde (FA) in 0.1 M phosphate buffer (PB) for 15 min. Eighty micrometer thick coronal sections from the dorsal hippocampus were cut, cryoprotected in 30% glycerol, and pieces from the CA1 area were frozen with a high-pressure freezing machine (HPM100, Leica Microsystems, Vienna, Austria) and fractured in a freeze-fracture machine (EM ACE900, Leica) as described in *Lorincz and Nusser, 2010*. Tissue debris were digested from the replicas with gentle stirring in a TBS solution containing 2.5% SDS and 20% sucrose (pH = 8.3) at 80°C for 18 hr. The replicas were then washed in Tris-buffered saline (TBS) containing 0.05% bovine serum albumin (BSA) and blocked with 5% BSA in TBS for 1 hr followed by an incubation in a solution of the following antibodies: rabbit polyclonal anti-Kv3.1b (1:1600; Synaptic Systems, SySy, Goettingen, Germany, Cat# 242 003, RRID:AB_11043175), rabbit polyclonal Munc13-1 (1:200, SySy, Cat# 126 103, RRID:AB_887733, raised against AA 3–317), a guinea pig polyclonal Munc13-1 (1:200, produced in collaboration with SySy against AA 364–469), and a guinea pig polyclonal PSD-95 (1: 500, SySy, Cat# 124 014, RRID:AB_2619800) antibody. In three experiments from four mice, the Munc13-1 antibody was mixed with a guinea pig Cav2.1 (1:3000, SySy, Cat# 152 205, RRID:AB_2619842) antibody, but only the Munc13-1 signal was analyzed in the present study. This was followed by an incubation in 5% BSA in TBS containing the following secondary antibodies: goat anti-rabbit IgGs (GAR) coupled with 5 nm or 10 nm gold particles (1:80 or 1:100; British Biocell International, BBI, Crumlin, UK) or donkey anti-guinea pig IgGs coupled with 12 nm gold particles (1:25, Jackson ImmunoResearch, Ely, UK) or goat anti-guinea pig IgGs coupled with 5 nm or 15 nm gold particles (1:100, BBI). Finally, replicas were rinsed in TBS and distilled water, before they were picked up on parallel bar copper grids and examined with a Jeol1011 EM (Jeol, Tokyo, Japan). The rabbit Munc13-1 antibody was raised against an intracellular epitope, resulting in a labeling on the protoplasmic face (P-face); therefore, nonspecific labeling was determined on surrounding exoplasmic-face (E-face) plasma membranes and was found to be $5.7 \pm 0.8$ gold particle/$\mu m^2$.

To quantify the Munc13-1 densities in the AZs of axon terminals targeting Kv3.1b + dendrites and somata, all experiments were performed using the 'mirror replica method' (*Éltes et al., 2017*; *Hagiwara et al., 2005*). With this method, replicas are generated from both matching sides of the fractured tissue surface, allowing the examination of the corresponding E- and P-faces of the same membranes. The AZs were delineated on the P-face based on the underlying high density of intramembrane particles.

To test the variability in the density of gold particles due to the stochastic binding of Abs to their epitopes, we performed the following modeling. We assume that Ab binding can be approximated with a binomial process where the probability of an Ab binding to an epitope is $p$ and the total number of epitopes is $N_{ep}$. First, we assumed a $p$ of 0.5 and an $N_{ep}$ of 80, to model the average of 40 gold particles per synapse for our PSD-95 labeling. The CV of this binomial distribution was 11% (mean = 40; SD = 4.4), which is very close to that of our experimental data for PSD-95, indicating a potential labeling efficiency of ~50% and 80 PSD molecules. Then we tested the effect of lowering the labeling efficiency an order of magnitude (p=0.05) and therefore increasing the $N_{ep}$ by a factor of 10. The CV (15.2%) of the resulting binomial distribution was indeed larger but was still about half of that we obtained for Munc13-1 experimentally. Another order of magnitude decrease in $p$ is still inconsistent with our experimental data, where the CV remains virtually the same (CV = 15.5%), but the $N_{ep}$ exceeds the number of protein present in a synapse. In summary, our modeling demonstrates that all variance (CV ~10%) in the density of PSD-95 immunolabeling could originate from a random process of Ab binding with a $p$ of 0.5, but such process is responsible for no more than 25% of the variance (CV ~15% out of the 33% experimental data) for the Munc13-1 labeling.

## Analysis of the distribution of Munc13-1 protein within the AZs

We used a Python-based open-source software with a graphical user interface, GoldExt (*Szoboszlay et al., 2017*; available on the website: http://www.nusserlab.hu/) to analyze gold particle distributions. Coordinates of the immunogold particles and corresponding AZ perimeters were extracted from EM images. Spatial organization of immunogold particles in presynaptic AZs was analyzed on the population of AZs using mean nearest-neighbor distance (NND) and a Ripley analysis (*Rebola et al., 2019*; *Ripley, 1979*). For the NND analysis, we calculated the mean of the NNDs of all gold particles within an AZ and that of random distributed gold particles within the same AZ (same number of gold particles, 200 repetitions). The NNDs were then compared statistically using the Wilcoxon signed-rank test. We used a variance stabilized and boundary corrected version of the Ripley's K function, called H-function (Hr) to examine whether particle distributions within individual AZs are clustered or dispersed over a range of spatial scales according to *Rebola et al., 2019*. To determine the number of clusters in Munc13-1 labeled AZs, we used the density-based clustering algorithm, DBSCAN (*Ester et al., 1996*).

## In vitro electrophysiology

### Slice preparation

Acute 300 µm thick coronal dorsal hippocampal slices were cut from C57Bl6/J (Jackson Laboratories, Bar Harbor, ME) (n = 70), Tg(Chrna2-Cre)OE25Gsat/Mmucd (RRID:MMRRC_036502-UCD, on C57Bl6/J background) (n = 18), sst $^{tm3.1\ (flop)\ Zjh}$/J (RRID:Cat_JAX:028579, RRID:IMSR_JAX:028579 on C57Bl6/J background) (n = 2), and Tg(Vipr2-cre)KE2Gsat/Mmucd (RRIP: MMRRC_034281-UCD) × Dlx5/6-Flpe (Tg(mI56i-flpe)39Fsh/J), (RRID:IMSR_JAX:010815) on C57Bl6/J background (n = 1) mice of both sexes (postnatal day 52–86). Animals were anaesthetized with a ketamine, xylasine, pypolphene cocktail (0.625, 6.25, 1.25 mg/ml, respectively, 10 µl/g body weight) and then decapitated or perfused with ice-cold cutting solution containing (in mM): sucrose, 205.2; KCl, 2.5; NaHCO$_3$, 26; CaCl$_2$, 0.5; MgCl$_2$, 5; NaH$_2$PO$_4$, 1.25; and glucose, 10, bubbled with 95% O$_2$ and 5% CO$_2$. The brain was quickly removed into ice-cold cutting solution, and coronal slices containing the dorsal hippocampus were cut using a Leica vibratome (VT1200S, Leica, Wetzlar, Germany) and placed in a submerged-type chamber in ACSF containing (in mM): NaCl, 126; KCl, 2.5; NaHCO$_3$, 26; CaCl$_2$, 2; MgCl$_2$, 2; NaH$_2$PO$_4$, 1.25; glucose, 10 saturated with 95% O$_2$ and 5% CO$_2$ (pH = 7.2–7.4) at 36°C, which was then gradually cooled down to 22–24°C. Recordings were carried out in the same ACSF 32–33°C, slices were kept up to 6 hr.

## Electrophysiology and data analysis

Patch pipettes were pulled (Zeitz Universal Puller; Zeitz-Instrumente Vertriebs, Munich, Germany) from thick-walled borosilicate glass capillaries with an inner filament (1.5 mm outer diameter, 0.86 mm inner diameter; Sutter Instruments, Novato, CA). Pipette resistance was 4–5 MΩ when filled with the intracellular solution containing (in mM): K-gluconate, 130; KCl, 5; MgCl$_2$, 2; EGTA, 0.05; creatine phosphate, 10; HEPES, 10; ATP, 2; GTP, 1; biocytin, 7; glutamate, 20 (for presynaptic PCs only) (pH = 7.3; 290–300 mOsm). All recordings were carried out in the presence of 0.35 mM γ-DGG (Tocris, Bristol, UK; #112) and 2 µM AM251 (Tocris; #1117). All drugs were applied using a recirculating system with a peristaltic pump (3–5 ml/min). All drugs were ordered from Sigma (St. Luis, MO), unless indicated otherwise.

Recordings were obtained using either a Multiclamp 700A or 700B amplifier (Molecular devices, CA), and signals were filtered at 6 kHz (Bessel filter) and digitized at 50 kHz with DigiData 1550A AD converter (Molecular Devices, San Jose, CA). Data were collected and analyzed using pClamp10_7 software (Molecular Devices, CA). Cell pairs where the access resistance of the postsynaptic IN exceeded 25 MΩ, the PCs access resistance exceeded 35 MΩ or the access change was >20% were excluded from the study. Cells were visualized using infrared differential interference contrast (DIC) method using an Olympus BX51 microscope with a 40× water immersion objective (NA = 0.8) or Nikon Eclipse FN1 microscope (Nikon, Tokyo, Japan) with a 40× water immersion objective (NA = 0.8).

Both PCs and FSINs in the hippocampal CA1 area were identified by their position and shape and size of the somata in the DIC image. INs were held at −65 mV in current-clamp mode, and firing properties were determined from their responses to square current injections (500 ms, from −300

pA to +300, 50 pA steps). Neurons with a narrow spike width, producing high-frequency spiking in response to large depolarizing current injections and displaying lack of a sag in response to hyper-polarizing current injections were considered FSINs in accordance with the literature. Presynaptic CA1 PCs were held at −65 mV in current-clamp mode and postsynaptic FSINs were held at −65 mV in voltage-clamp mode. In the presynaptic PCs, six APs were evoked at 40 Hz followed by a recovery pulse after 300 or 500 ms with 1.5 ms long 1.5 nA depolarizing current pulses, which was repeated in every 8 s. The measured EPSC amplitude values were corrected with the amplitude of the baseline negative peak. To investigate changes in quantal parameters, $Pv$ was increased by elevating extra-cellular $[Ca^{2+}]$ to 6 mM. MPFA was carried out according to *Biró et al., 2005*. If the variance for the largest mean value was the largest, the cell was excluded from the analysis. This criterium served to ensure that the $Pv$ is likely to be >0.5 increasing the reliability of deciphering the quantal parameters from the parabola fit. The mean and variance of EPSC peak amplitudes were calculated in 6 mM $[Ca^{2+}]$ recordings from 24 to 30 sweeps (contaminated sweeps were excluded, and if the total number of sweeps was <24, the cell pair was omitted from the analysis). Plots of mean versus variance values were fitted with a parabola to determine $N$ and $q$. $Pv$ was calculated as $P1/(N * q)$ where $P1$ is the peak amplitude of the first EPSC of the train. All electrophysiological data were analyzed with Microsoft Excel and OriginPro 2018 (OriginLab, Northampton, MA) as described above.

## Postembedding immunofluorescent reactions
### Tissue preparation
After recordings, slices were placed in a fixative containing 4% FA and 0.2% picric acid in 0.1 M PB (pH = 7.4) for 12 hr at 4°C. They then were embedded in agarose (2%) and re-sectioned at ~150 μm thickness. The biocytin-filled cells were visualized with Cy3-conjugated streptavidin (1:1000, Jackson ImmunoResearch, Bar Harbor, ME) in TBS containing 0.2% Triton X-100. Sections were then treated with uranyl acetate, dehydrated in a graded series of ethanol, incubated in acetonitrile, and flat-embedded in epoxy resin (Durcupan) as described in *Holderith et al., 2020*. Putative contacts between the recorded neurons were identified with visual inspection at high magnification (60×, 1.35 NA objective, Olympus FV1000 microscope, Tokyo, Japan). Tissue blocks containing the biocy-tin-filled processes were re-embedded, and ultrathin (70 or 200 nm) serial sections were cut and mounted on adhesive Superfrost Ultra plus slides. Potential contact sites between the presynaptic PC boutons, and the postsynaptic dendrites were identified on the ultrathin sections, imaged using a confocal microscope (Olympus FV1000), and reconstructed with a custom-made ImageJ plugin (HyperStackStitcher, 3DHistech, available on the website: http://www.nusserlab.hu/software.html).

## Postembedding immunofluorescent labeling
Etching of the resin, antigen retrieval, immunolabeling, and elution were carried out as reported previously (*Holderith et al., 2020*). Primary and secondary antibodies were the followings: rabbit poly-clonal Munc13-1 (1:200, SySy Cat# 126 103, RRID:AB_887733), guinea pig polyclonal PSD-95 (1:200, SySy, Cat# 124 014, RRID:AB_2619800), rabbit polyclonal vGluT1 (1:200, SySy, Cat# 135–302, RRID:AB_887877), guinea pig polyclonal pan-AMPAR (1:200, Frontier Cat# Af580, RRID:AB_257161), goat anti-rabbit IgGs coupled with Abberior635P (1:200, Abberior GmbH, Goettingen, Germany), goat anti-guinea pig IgGs coupled with Abberior635P (1:200), and donkey anti-rabbit coupled with Alexa488 (1:200, Jackson ImmunoResearch). After labeling, sections were washed and mounted in Slowfade Diamond Antifade Mountant (ThermoFisher Scientific, Waltham, MA). Images of all sections containing the identified synaptic contacts were taken at high magnification using an Abberior Instruments Expert Line STED microscope (100 × 1.4 NA objective on an Olympus BX63 microscope, Abberior Instruments GmbH, Goettingen, Germany). After imaging, immunoreagents were eluted and a new round of labeling was performed.

## Image analysis
A custom-made ImageJ plugin (HyperStackStitcher) was used to align images of ultrathin serial sections. To quantitatively analyze immunolabelings, ROIs were placed over the identified and surrounding synapses in ImageJ, and background subtracted integrated fluorescence intensities were measured. Signals of the identified synapses were normalized to the population mean calculated from 49 to 89 surrounding synapses.

To quantify the Munc13-1 and PSD-95 signal of random synapses located on FSIN dendrites, the measured integrated fluorescence intensities were normalized to the mean of the analyzed synapse population. Munc13-1 to PSD-95 ratios were calculated in each synapse, and the coefficient of variation (CV) of these ratios were assessed.

### Statistical analysis

Shapiro-Wilk test was used to test the normality of our data. To compare two dependent groups, paired t-test or Wilcoxon signed-rank test was used. Correlations were determined with Spearman's rank correlation and the regression coefficient ($r_s$); related p-values were calculated from two-tailed Student's t-distribution. In *Figure 3J*, the p-values were adjusted with the Holm–Bonferroni methods to account for the repeated use of the same data. Statistical tests were performed in Statistica (TIBCO Software Inc, Palo Alto, CA) or OriginPro 2018 (OriginLab).

## Acknowledgements

ZN is the recipient of a European Research Council Advanced Grant (ERC-AG 787157) and a Hungarian National Brain Research Program (NAP2.0) grant. The financial support from these funding bodies is gratefully acknowledged. We thank Éva Dobai and Dóra Rónaszéki for their excellent technical assistance.

## Additional information

### Funding

| Funder | Grant reference number | Author |
|---|---|---|
| European Research Council | ERC-AG 787157 | Zoltan Nusser |
| National Research Development and Innovation Office | NAP2.0 | Zoltan Nusser |

The funders had no role in study design, data collection and interpretation, or the decision to submit the work for publication.

### Author contributions

Maria Rita Karlocai, Andrea Lorincz, Conceptualization, Data curation, Formal analysis, Methodology; Judit Heredi, Data curation, Formal analysis, Methodology; Tünde Benedek, Data curation, Methodology; Noemi Holderith, Conceptualization, Formal analysis; Zoltan Nusser, Conceptualization, Formal analysis, Supervision, Funding acquisition, Investigation, Writing - original draft, Writing - review and editing

### Author ORCIDs

Noemi Holderith http://orcid.org/0000-0002-0024-3980
Andrea Lorincz https://orcid.org/0000-0003-2430-5290
Zoltan Nusser https://orcid.org/0000-0001-7004-4111

### Ethics

Animal experimentation: All the experiments were carried out according to the regulations of the Hungarian Act of Animal Care and Experimentation 40/2013 (II.14) and were reviewed and approved by the Animal Committee of the Institute of Experimental Medicine, Budapest.

### Decision letter and Author response

Decision letter https://doi.org/10.7554/eLife.67468.sa1
Author response https://doi.org/10.7554/eLife.67468.sa2

## Additional files

### Supplementary files
• Transparent reporting form

### Data availability
Source data have been provided for all figures.

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
