## [Decision Letter]

**Acceptance summary:**

Here the authors study how single synapses compute information by tuning the properties of the individual components that drive synaptic communication. Using cutting edge physiology and anatomical analysis, they show that the reliability of synaptic communication does not only depend on how many units drive synaptic communication, but also that individual units vary in their quantitative molecular composition. These results provide new insight in to the diversity of excitatory release sites that mediate communication between neurons.

**Decision letter after peer review:**

Thank you for submitting your article "Diversity of excitatory release sites" for consideration by *eLife*. Your article has been reviewed by 3 peer reviewers, one of whom is a member of our Board of Reviewing Editors, and the evaluation has been overseen by Gary Westbrook as the Senior Editor. The following individual involved in review of your submission has agreed to reveal their identity: Christian Rosenmund (Reviewer #2). The reviewers have discussed their reviews with one another, and the Reviewing Editor has drafted this to help you prepare a revised submission.

Essential revisions:

Using a combination of powerful approaches, the authors demonstrate large variability in the number of release sites at hippocampal excitatory synapses onto fast spiking interneurons. They report that individual synapses showed highly variable amounts of Munc13-1within the active zones (AZs) that have the same number of release sites, as well as a synapse size-independent variability in the number of Munc13-1 clusters per AZ and the Munc13-1 content of individual clusters. The reviewers were enthusiastic about the question and the depth of analysis using multiple sophisticated approaches, and found the conclusions to be largely supported by the data. However, there were mixed views about whether there was significant new insight into molecular heterogeneity of release sites relative to existing knowledge, and some conceptual and presentation issues that reduced enthusiasm. These will be important to address in the revised manuscript. Please also consider the comments of the reviewers in their original reviews below.

1. Conceptual presentation of manuscript. The authors' data does not provide evidence that Munc-13 variability is the primary cause of release heterogeneity. Such implications would require additional evidence that addresses functional consequences of Munc-13 variability. Framing the question addressed by the current dataset more narrowly and/or additional data, may clarify the significance of the current results.

2. Interpretation of Munc13 variability. More explicit interpretation of current results and integration into existing literature of release site heterogeneity and Munc13 biology is needed. The authors need to establish the main conceptual advance, which should be reflected in the title.

*Reviewer #1 (Recommendations for the authors):*

The functional and molecular analysis of CA1 to fast-spiking interneuron glutamatergic synapses is clearly described and convincing. I do not have technical concerns. My main concern relates to the significance of the interpretation and how it relates to the function of Munc13 and release sites.

Specifically, the manuscript focuses on measuring Munc13 at active zones, with the rationale that it is the core component of release sites (Sakamoto et al.). It seems that the variability in Munc13 cluster density in AZs and Munc13 molecules per cluster is thus unexpected. Beyond potentially ruling out the hypothesis that presynaptic release sites are constructed from molecularly uniform release sites, the authors should clarify the significance and implications of this variability. Are the authors proposing that release sites are composed of Munc13 nanoassemblies with different diameters depending on the Munc13 content? Would other proteins such as RIM1 likewise vary with Munc13 and would there be functional differences expected from the differing composition? Explicitly addressing how these results change the current view of release site composition and function in the abstract and introduction (and perhaps title) would establish the significance of the results.

*Reviewer #2 (Recommendations for the authors):*

The finding that the post synapse as defined by PSD95 labeling was much less variable, indicates that pre- and postsynaptic makes do not necessarily correlate, arguing somewhat against the transsynaptic nano column concept as a main organizing principles. Thus, pre and post synapses are only loosely linked in their composition and function. I think this issue deserves a bit more discussion, in contrast to the discussion of the role of multivesicular release (MVR) which is less close to what the authors actually have analyzed in this work.

*Reviewer #3 (Recommendations for the authors):*

1. Logical fallacy in approach/discussion. The authors ask the question of "what the molecular correlates of the variability in the number of RSs within individual presynaptic AZs are". The fact that authors choose to detect Munc-13 does not mean that it is the primary cause of variability, or even that it is the cause or a consequence. The manuscript should be re-written to address this logical issue.

The authors made a related point themselves in Discussion: "The priming efficacy of Munc13-1 depends on its interaction with RIM and RIM binding protein (Brockmann et al., 2020) therefore predicting the functional consequence of the different amounts of Munc13-1 per RS might require the determination of these molecules in individual Munc13-1 clusters".

2. Terminology is used loosely and creates confusion. There are multiple places where authors refer to the number of release sites as N, N/AZ, or "the number of functional RSs per AZ". Also different values are given in different parts of the paper without explanation. While these values are all correct and complementary, much more clarity in the description is need to be able to navigate the results. For example:

"found large variability in N (9.9 {plus minus} 9.0, CV = 0.91; Figure 2B)".

"in the number of RSs within individual AZs (ranges from 1 to 17, CV = 0.75)".

"found an average of 5.4 {plus minus} 2.5 clusters per AZ".

"the number of functional RSs per AZ (4.9 {plus minus} 3.7)".

"the N / AZ (4.9 {plus minus} 3.7, CV = 0.75, from 1 to 17)".

There is also another similar term: rs = 0.37; Figure 2H, which has nothing to do with RS, but can cause some confusion for the readers.

Moreover, in Discussion: "By determining the number of RSs with quantal analysis….". Authors made a point themselves that quantal analysis gives N, not RS number because the number of connections is larger than 1; I quote the manuscript: "Because PC – FSIN connections are not mediated by single synapses the overall variability in N is not simply the consequence of different Ns per AZs, but also the function of the number of synaptic contacts formed by the presynaptic axon on the postsynaptic cell".

3. The title needs to reflect some specifics. The current title sounds like a review title.

4. Interpretation of immuno gold data. If I understand the figure correctly, this data does not give distance between clusters, but distance between the gold particles. Release sites are unlikely to be ~25nm apart as authors imply, because vesicles are 40-50nm in diameter and release site occupancy is close to 1 at rest, as the authors pointed out. Therefore I would expect a minimum of ~40-50nm between the clusters, but realistically even more given some uncertainty in position determination.

5. Some key references to previous findings are missing. Analysis of Munc13 clusters per AZ has been initially shown in Tang et al., Nature 2016. The finding of a large variability in RS number per AZ has been initially reported in Maschi et al., Neuron 2017. There are two most relevant earlier papers that surprisingly are omitted. These papers need to be cited and discussed.

6. Is it possibility that results of Figure 5 could be a result of different efficiency of the two antibodies because both co-vary similarly with respect to the size of the AZ but Munc13 is simply more "noisy". If so, a control using different Abs would seem to be appropriate as this is a major point.

[Editors' note: further revisions were suggested prior to acceptance, as described below.]

Thank you for submitting your article "Diversity of excitatory release sites" for consideration by *eLife*. Your resubmitted article has been overseen by a Reviewing Editor and Gary Westbrook as the Senior Editor. The Reviewing Editor has drafted these comments to help you prepare a final revised submission.

Essential revisions:

Please expand on your responses to the reviewer concerns more fully within the manuscript. For example:

1. Explaining the significance of assessing Munc-13 in the abstract or introduction would clarify the results for a general audience (Rev 1).

2. Please include in the discussion how their results fit into the nano column concept as a main organizing principle (Rev 2).

3. In addition to explaining the logic of assessing Munc13 and an explicit acknowledgement that the consequences of Munc13 variability are not known, the authors should provide a title that is more informative of the significance of the work (Rev 3).

---

## [Author Response]

Reviewer #1 (Recommendations for the authors):The functional and molecular analysis of CA1 to fast-spiking interneuron glutamatergic synapses is clearly described and convincing. I do not have technical concerns. My main concern relates to the significance of the interpretation and how it relates to the function of Munc13 and release sites.

We were delighted to read that the reviewer has found our data convincing and that she/he had no technical concern. We will do our best to clarify uncertainties regarding our interpretation of our data.

Specifically, the manuscript focuses on measuring Munc13 at active zones, with the rationale that it is the core component of release sites (Sakamoto et al.). It seems that the variability in Munc13 cluster density in AZs and Munc13 molecules per cluster is thus unexpected. Beyond potentially ruling out the hypothesis that presynaptic release sites are constructed from molecularly uniform release sites, the authors should clarify the significance and implications of this variability. Are the authors proposing that release sites are composed of Munc13 nanoassemblies with different diameters depending on the Munc13 content?

Yes, this is probably the most likely interpretation of our data (large variability in Munc13-1 molecules per clusters). However, the functional consequence of our data is yet unknown. Methods similar to that used by Maschi and Klyachko can reveal the variability in Pv among release sites within individual AZs. Unfortunately, such superresolution methods so far have only been possible in cultured neurons.

Would other proteins such as RIM1 likewise vary with Munc13 and would there be functional differences expected from the differing composition?

As stated in the Discussion, knowing the amounts of RIM and RIM-BP in the Munc13-1 clusters would be brilliant. Unfortunately, the currently available antibodies to Rim1/2 gave very faint, but specific looking signal, but no signal could be obtained against the RIM-BP in our post-embedding reactions. Thus, there is technical reasons why they are not included in the manuscript.

Explicitly addressing how these results change the current view of release site composition and function in the abstract and introduction (and perhaps title) would establish the significance of the results.Reviewer #2 (Recommendations for the authors):The finding that the post synapse as defined by PSD95 labeling was much less variable, indicates that pre- and postsynaptic makes do not necessarily correlate, arguing somewhat against the transsynaptic nano column concept as a main organizing principles. Thus, pre and post synapses are only loosely linked in their composition and function. I think this issue deserves a bit more discussion, in contrast to the discussion of the role of multivesicular release (MVR) which is less close to what the authors actually have analyzed in this work.

We appreciate the positive comments of the reviewer and acknowledge his view that our results regarding the nano-column organization of the synapse worth discussing in more details. We are fully aware that our results do not support this, now popular, concept, but this is not the first paper from our laboratory that contradicts with that. We have shown (Szoboszlay et al., Sci Rep 2017) that synapses are heterogeneous regarding the subsynaptic distribution of AMPARs; in some synapses it has a distribution that is not different from random and in some others, it shows a distribution that is consistent with a regular pattern. None of these supports the nanocolumn model! We clearly cannot disprove data obtained in one specific synapse but can show that it is not a general organizational principle of central glutamatergic synapses (more of an exception rather than the rule).

Reviewer #3 (Recommendations for the authors):1. Logical fallacy in approach/discussion. The authors ask the question of "what the molecular correlates of the variability in the number of RSs within individual presynaptic AZs are". The fact that authors choose to detect Munc-13 does not mean that it is the primary cause of variability, or even that it is the cause or a consequence. The manuscript should be re-written to address this logical issue.

The reviewer is correct that it is not an unquestionable fact that Munc13-1 is a key component of the release site, but most recent papers point toward this molecule as a key component. Therefore, we decided to use this as a molecular marker. And indeed, we do not know whether the variability in its amounts causes changes in docking site occupancy, priming state, or release probability, but we believe that our data demonstrate for the first time that RSs are heterogeneous with respect to their molecular content (and indeed, with currently unknown consequences).

The authors made a related point themselves in Discussion: "The priming efficacy of Munc13-1 depends on its interaction with RIM and RIM binding protein (Brockmann et al., 2020) therefore predicting the functional consequence of the different amounts of Munc13-1 per RS might require the determination of these molecules in individual Munc13-1 clusters".

Indeed, the reviewer is right that in the ideal world, we should have correlated the *N*/AZ in a four-dimensional space with the amounts of Munc13-1, Rim1/2 and RIM-BP (or even in a higher dimensional space with many other key pre- and postsynaptic proteins). Unfortunately, the available antibodies to Rim1/2 gave very faint, but specific looking signal, but no signal could be obtained against the RIM-BP in our post-embedding reactions. Accepting these limitations, the best we could do was to demonstrate for the first time that AZs with identical *N* have largely variable amounts of Munc13-1 and only mention in the discussion that future investigations with Rim and RIM-BP will, for sure, unravel the genuine complexity in the molecular organization of the RSs.

2. Terminology is used loosely and creates confusion. There are multiple places where authors refer to the number of release sites as N, N/AZ, or "the number of functional RSs per AZ". Also different values are given in different parts of the paper without explanation. While these values are all correct and complementary, much more clarity in the description is need to be able to navigate the results. For example:"found large variability in N (9.9 {plus minus} 9.0, CV = 0.91; Figure 2B)"."in the number of RSs within individual AZs (ranges from 1 to 17, CV = 0.75)"."found an average of 5.4 {plus minus} 2.5 clusters per AZ"."the number of functional RSs per AZ (4.9 {plus minus} 3.7)"."the N / AZ (4.9 {plus minus} 3.7, CV = 0.75, from 1 to 17)".

We regret that we were not consistent with our nomenclature. Now, we have changed this in the revised manuscript and refer to *N* as the number of release sites between two connected cells; and *N*/AZ as the number of release sites within individual active zones.

There is also another similar term: rs = 0.37; Figure 2H, which has nothing to do with RS, but can cause some confusion for the readers.

Indeed, our abbreviation of the Spearman regression coefficient (r_s_) has the same letters as that of release sites (RS), but one is written with capital letters and the other without and the ‘s’ in subscript. We hope that they are distinct enough not to cause confusion. Now we clearly state the abbreviations in the text to prevent potential confusion.

Moreover, in Discussion: "By determining the number of RSs with quantal analysis….". Authors made a point themselves that quantal analysis gives N, not RS number because the number of connections is larger than 1; I quote the manuscript: "Because PC – FSIN connections are not mediated by single synapses the overall variability in N is not simply the consequence of different Ns per AZs, but also the function of the number of synaptic contacts formed by the presynaptic axon on the postsynaptic cell".

We would like to clarify that the *N*, as obtained from quantal analysis refers to all release sites. If a connection is mediated by multiple synapses (contacts) then the *N* is not the number of RS per synapse, but the total number of RS in all synapses. Therefore, we must know the number of synapses (synaptic contacts) in order to calculate the *N*/AZ, which is the relevant measure when its relationship with the amount of synaptic proteins is concerned.

3. The title needs to reflect some specifics. The current title sounds like a review title.

We agree that our choice of title is a bit unusual, and we have changed it acccording the suggestion..

4. Interpretation of immuno gold data. If I understand the figure correctly, this data does not give distance between clusters, but distance between the gold particles. Release sites are unlikely to be ~25nm apart as authors imply, because vesicles are 40-50nm in diameter and release site occupancy is close to 1 at rest, as the authors pointed out. Therefore I would expect a minimum of ~40-50nm between the clusters, but realistically even more given some uncertainty in position determination.

Yes, the reviewer is correct that we measured the nearest neighbor distances of each gold particle that has a median value of ~25 nm. We use this to address whether the gold particles have a within-AZ distribution that is significantly different from random or not. We would like to point out that we have not provided the inter-cluster distance in the original manuscript. Now, we have measured the inter-cluster distances, which has a mean of 178+-93 nm (n = 1593, distances from 102 AZs). However, a functionally more relevant measure is the nearest neighbor inter-cluster distance, which tells us how close the immediate neighbors are, but ignores long, second and third neighbor distances. This value has a mean of 85 +-34 nm (n = 568 distances from 102 AZs) that is consistent with the spacing and size of docked synaptic vesicles.

5. Some key references to previous findings are missing. Analysis of Munc13 clusters per AZ has been initially shown in Tang et al., Nature 2016. The finding of a large variability in RS number per AZ has been initially reported in Maschi et al., Neuron 2017. There are two most relevant earlier papers that surprisingly are omitted. These papers need to be cited and discussed.

We regret to leave out these two references from our original manuscript. Tang et al. mainly concentrate on Rim1/2 and only briefly mentions Munc13-1 in the manuscript. Regarding Maschi et al., 2017, we heavily cited their follow up manuscript that contains all relevant data from this group. However, in accordance with the suggestion of the reviewer, now we included both in our revised manuscript.

6. Is it possibility that results of Figure 5 could be a result of different efficiency of the two antibodies because both co-vary similarly with respect to the size of the AZ but Munc13 is simply more "noisy". If so, a control using different Abs would seem to be appropriate as this is a major point.

The reviewer is correct in pointing out that the normalized variability depends on the *p* and *N* in a binomial distribution. To quantitatively test this, we have performed the following modeling. We assume that Ab binding can be approximated with a binomial process where the probability of an Ab binding to an epitope is *p* and the total number of epitopes is *N*. First, we assumed a *p* of 0.5 and an *N* of 80, to model the average of ~40 gold particles per synapse for the PSD-95 labelling. We then calculated the mean (40) and SD (4.4) of the binomial distribution, which resulted in a CV of 11%. That is very close to that found experimentally for PSD-95 density, indicating a potential labeling efficiency of ~50% and 80 PSD molecules. Then we tested the effect of lowering the labelling efficiency an order of magnitude (p = 0.05) and therefore increasing the *N* by a factor of 10. The CV (15.2%) of the resulting binomial distribution was indeed larger but was still about half of that we obtained for Munc13-1 experimentally. Would another order of magnitude decrease in *p* be consistent with the experimental data? No. In such a case, the CV remains virtually the same (CV = 15.5%) and the *N* (8000) way exceeds the number of Munc13-1 proteins present in a synapse. In summary, our modelling (which is now included in the manuscript) demonstrates that all variance (CV ~10%) in our PSD-95 immunolabeling could originate from the random process of Ab binding with a probability of about 50%, but such process is responsible for no more than 25% of the variance (CV ~15% out of the 33% experimental data) for the Munc13-1 labeling.

[Editors' note: further revisions were suggested prior to acceptance, as described below.]

Essential revisions:Please expand on your responses to the reviewer concerns more fully within the manuscript. For example:1. Explaining the significance of assessing Munc-13 in the abstract or introduction would clarify the results for a general audience (Rev 1).

We have added the following text in the abstract:

“High resolution molecular analysis of functionally characterized synapses reveals variability in the content of one of the key vesicle priming factors – Munc13-1 – of AZs that possess the same *N*.”

and in the Introduction:

“Since Munc13-1 and its invertebrate homologues are essential for docking and priming of synaptic vesicles (Augustin et al., 1999; Imig et al., 2014; Weimer et al., 2006) and the fact that its cluster numbers show tight correlation with *N* suggests that it is a key molecule of the RSs and can be used as its molecular marker.”

2. Please include in the discussion how their results fit into the nano column concept as a main organizing principle (Rev 2).

We have now added a new paragraph to the discussion:

“A recent study applied superresolution light microscopy to examine the spatial relationship between pre- and postsynaptic molecules and came to the conclusion that key molecules are arranged in transsynaptic nanocolumns (Tang et al., 2016).[…] All these data taken together indicates that the nanocolumn arrangement of pre- and postsynaptic key signalling molecules is not a universal feature of all glutamatergic synapses.”

3. In addition to explaining the logic of assessing Munc13 and an explicit acknowledgement that the consequences of Munc13 variability are not known, the authors should provide a title that is more informative of the significance of the work (Rev 3).

We have changed our title now to:

**“Variability in the Munc13-1 content of excitatory release sites.”**

We hope that our new title is more informative regarding what aspect of release site variability our data remonstrate.

In addition, now we explicitly state at the end of the Introduction as well that:

‘Whether vesicles that are docked to RSs with different amounts of Munc13-1 have distinct *Pv* or not, remains to be seen.’